# Glioma-neuronal circuit remodeling induces regional immunosuppression

Takahide Nejo [1], Saritha Krishna[1], Akane Yamamichi[1], Senthilnath Lakshmanachetty[1], Christian Jimenez[1], Kevin Y. Lee[1], Donovan L. Baker [1], Jacob S. Young[1], Tiffany Chen[1], Su Su Sabai Phyu[1], Lan Phung[1], Marco Gallus[1,2], Gabriella C. Maldonado [1], Kaori Okada[1], Hirokazu Ogino [1], Payal B. Watchmaker[1], David Diebold[1], Abrar Choudhury [1,3,4], Andy G. S. Daniel[1], Cathryn R. Cadwell [1,4,5,6], David R. Raleigh [1,3,4], Shawn L. Hervey-Jumper [1,5] ✉ & Hideho Okada [1,7] ✉

Neuronal activity-driven mechanisms influence glioblastoma cell proliferation and invasion, while glioblastoma remodels neuronal circuits. Although a subpopulation of malignant cells enhances neuronal connectivity, their impact on the immune system remains unclear. Here, we show that glioblastoma regions with enhanced neuronal connectivity exhibit regional immunosuppression, characterized by distinct immune cell compositions and the enrichment of anti-inflammatory tumor-associated macrophages (TAMs). In preclinical models, knockout of Thrombospondin-1 (TSP1/*Thbs1*) in glioblastoma cells suppresses synaptogenesis and glutamatergic neuronal hyperexcitability. Furthermore, TSP1 knockout restores antigen presentation-related genes, promotes the infiltration of pro-inflammatory TAMs and CD8 + T-cells in the tumor, and alleviates TAM-mediated T-cell suppression. Pharmacological inhibition of glutamatergic signaling also shifts TAMs toward a less immunosuppressive state, prolongs survival in mice, and shows the potential to enhance the efficacy of immune cell-based therapy. These findings confirm that glioma-neuronal circuit remodeling is strongly linked with regional immunosuppression and suggest that targeting glioma-neuron-immune crosstalk could provide avenues for immunotherapy.

Despite advances in the surgical and medical treatments for glioblastoma, the most common and aggressive malignant primary brain neoplasm, patients still face dismal prognoses[1]. Recent advancements have shed light on a previously unrecognized mechanism whereby neuronal activity drives glioblastoma growth[2–6] and invasion[7–9] through direct synaptic connections between neurons and glioblastoma cells[4,7], as well as paracrine growth factors from glioblastoma cells and excitatory neurons[2,3,5,6]. Conversely, glioblastoma cells induce neuronal hyperexcitability and neuronal circuit hypersynchrony[10–13].

The amount of functional connectivity between glioblastoma cells and the normal brain circuits negatively impacts patient survival through the tumor-derived synaptogenic factor thrombospondin-1 (TSP1, encoded by the *THBS1* gene)[14,15]. Furthermore, patient-derived glioblastoma cells from functionally connected intratumoral regions

[1]Department of Neurological Surgery, University of California, San Francisco, San Francisco, CA, USA. [2]Department of Neurosurgery, University Hospital Muenster, Muenster, Germany. [3]Department of Radiation Oncology, University of California, San Francisco, San Francisco, CA, USA. [4]Department of Pathology, University of California, San Francisco, San Francisco, CA, USA. [5]Weill Neurohub, San Francisco, CA, USA. [6]Kavli Institute for Fundamental Neuroscience, University of California, San Francisco, San Francisco, CA, USA. [7]Parker Institute for Cancer Immunotherapy, San Francisco, CA, USA. ✉e-mail: shawn.hervey-jumper@ucsf.edu; hideho.okada@ucsf.edu

are characterized by a proliferative and invasive phenotype in the presence of neurons. The fundamental discovery that the tumor microenvironment regulates malignant growth in an activity-dependent manner raises questions about additional cellular factors that may alter or drive glioblastoma proliferation. Intriguingly, single-cell RNA-sequencing analysis of patient tumor samples has revealed that TSP1 is predominantly expressed by glioblastoma cells within highly functionally connected (termed HFC) intratumoral regions, in addition to astrocytes and myeloid cells. In contrast, astrocytes emerge as the primary source of TSP1 expression in the lowly functionally connected (termed LFC) regions[14]. This observation suggests distinct gene expression programs between HFC and LFC intratumoral regions. Myeloid cells, represented by microglia, emerge in early development, respond to the local environment by altering their molecular and phenotypic states, and regulate neuronal activity[16,17]. Moreover, glioblastoma cells and tumor-associated macrophages (TAMs) engage in bidirectional crosstalk, where glioblastoma cells attract TAMs and the TAMs promote glioblastoma cell proliferation and invasion[18,19], preventing T-cell-mediated immune attack[20]. Therefore, activity-dependent glioblastoma proliferation may be governed by crosstalk between neurons, immune cells, and glioblastoma cells, which remains poorly understood.

This knowledge gap is paramount to address, as gaining a deeper understanding of immune-modulating mechanisms can unlock new therapeutic opportunities. Despite extensive efforts, and unlike the clinical success observed in various other malignancies over the past decade, cancer immunotherapy has yet to demonstrate efficacy for patients with glioblastoma[21,22]. Thus, identification of immune-modulating factors unique to the central nervous system could provide valuable insights for developing distinctive approaches and pave the way for successful immunotherapeutic interventions.

In this work, we demonstrate that glioma-neuronal circuit remodeling is strongly linked with regional immunosuppression and identify glioblastoma cell-derived TSP1 as a causal mediator of the suppressive immune microenvironment in the context of distinct patterns of glioma-neuronal circuit remodeling and neuronal activity. Furthermore, we show that pharmacological inhibition of glutamatergic excitatory signaling using an FDA-approved drug reprograms the immune microenvironment of intracerebral glioblastoma, specifically toward a less anti-inflammatory phenotype of TAMs, resulting in prolonged survival.

## Results
### Functionally connected intratumoral regions have distinct immunological programs
We aimed to better understand the distinctive transcriptional programs between tumor and immune cells from HFC and LFC regions characterized by intratumoral connectivity and local neuronal activity. Therefore, we re-analyzed the previously reported single-cell RNA-seq (sc-RNA-seq) datasets of clinical samples, in which annotations as either HFC- or LFC-derived had been assigned based on presurgical MEG and MRI imaging analyses (Supplementary Fig. 1a)[14]. We performed differential gene expression (DGE) analyses followed by gene set enrichment analyses (GSEA). The unbiased testing of 50 pathways from the MSigDB Hallmark collection revealed that numerous immune-related gene signatures were among the most significantly downregulated pathways in HFC compared with LFC regions (Supplementary Data 1). These downregulated pathways included *inflammatory response*, *interferon*-α and *-γ responses*, and *TNFα signaling* via *NFκB pathway* (Fig. 1a–c). These findings were consistently observed across multiple cell populations—tumor, myeloid, and lymphoid cells—while not in astrocytes (Supplementary Fig. 1b). In contrast, the glycolysis pathway was consistently upregulated in HFC areas across all three cell types (Fig. 1a–c), suggesting enhanced local neuronal activity and glucose consumption within these intratumoral regions[23,24].

Signature scoring analyses, both at the single-cell level and through pseudo-bulk analysis, corroborated that the pathways of the *inflammatory response*, *interferon-γ response*, and *TNFα signaling* via *NFκB* tend to be downregulated in HFC region cells compared to LFC region cells across the cell types in common (Supplementary Fig. 2). Demonstrative genes consistently expressed in poorly connected LFC regions (over HFC) included *CCL2/4, CD83, IL1B, ISG15, NFKB1*, and *STAT1* (Supplementary Fig. 1c–h). Moreover, GSEA of the *inflammatory response*, *interferon-γ response*, and *TNFα signaling via NFκB pathways* identified *CCL2* and *NFKB1* as the most common leading-edge genes across the three cell types, potentially playing central roles in distinguishing the immunological status of HFC and LFC regions (Supplementary Fig. 3). These findings led us to hypothesize that glioblastoma-intrinsic functional connectivity, which promotes the malignant behavior of glioblastoma, is associated with immune regulatory programs involving multiple cell types within these regions. Therefore, we aimed to investigate evidence of the relationship between the immune system and neuronal activity in glioblastoma.

### Myeloid cell populations are immunosuppressive within functionally connected intratumoral regions
Myeloid cells are a major component of the glioblastoma microenvironment[25], represented by TAMs originating from brain-resident microglia (Mg-TAMs) or bone-marrow monocyte-derived macrophages (Mo-TAMs)[26–28]. TAMs are typically polarized toward alternatively activated, anti-inflammatory phenotypes due to tumor-extrinsic factors, such as TGF-β, IL-10, GM-CSF, and CSF-1[29,30], although recent studies have also indicated that TAMs in glioblastoma can exist in an immature state without showing clear polarization[19,26]. Notably, recent single-cell studies have demonstrated that, although TAMs can simultaneously express canonical pro-inflammatory (also described as "classically activated" or "M1") and anti-inflammatory ("alternatively activated" or "M2") markers on the same cells, Mo-TAMs more frequently up-regulate anti-inflammatory cytokines compared to Mg-TAMs[26]. Therefore, while their oversimplified classification remains controversial[25], evaluating the composition of pro-inflammatory and anti-inflammatory TAM phenotypes and factors influencing the phenotype can still help us better understand the characteristics of the tumor microenvironment[26,27]. Furthermore, microglia activation regulates the activity of neurons, suggesting a potential role in glioblastoma proliferation[31].

To explore the characteristics of TAMs derived from HFC and LFC intratumoral regions, we investigated the myeloid cell populations (n = 3775 cells) extracted from the sc-RNA-seq dataset (Fig. 1d). Unsupervised clustering identified several distinct myeloid cell subclusters, including those of pro-inflammatory and anti-inflammatory status (Fig. 1e). Pro-inflammatory populations were represented by *CCL3, CD83, IL1B,* and *TNF*, while anti-inflammatory populations were represented by *CD163, LGALS1/3, LYZ, RNASE1, TGFBI,* and *VIM* (Supplementary Fig. 4a). In this analysis, myeloid cells from the HFC regions revealed trends toward an increase in anti-inflammatory populations (37.7% vs. 22.1% in LFC) and a decrease in pro-inflammatory populations (41.4% vs. 53.1% in LFC) compared with those from the LFC regions (Fisher's exact test $p = 6.1 \times 10^{-18}$) (Fig. 1f and Supplementary Fig. 4b–d). In addition to inflammatory status, we sought to discriminate between Mg- and Mo-TAMs, given the prior observation that the compositions of innate immunity populations dynamically change along with disease progression and aggressiveness[32,33]. Signature scoring analysis demonstrated that the distributions of Mg-TAMs and Mo-TAMs overlapped with those of pro-inflammatory and anti-inflammatory TAM populations, respectively (Fig. 1g). As such, myeloid cells from the HFC regions were estimated to have greater Mo-TAM populations (73.4% [in HFC] vs. 52.3% [in LFC]) and less Mg-TAM populations (20.9% vs. 37.6%) compared with their LFC counterparts (Fisher's exact test $p = 4.9 \times 10^{-25}$) (Fig. 1h and Supplementary Fig. 4e–g).

Moreover, given previous observations of TSP1 expression in HFC region myeloid cells[14], we investigated *TSP1* expression within this cell population. Among the entire myeloid cell population, 4.9% of the cells were identified as *TSP1*-positive (Supplementary Fig. 5a). In each

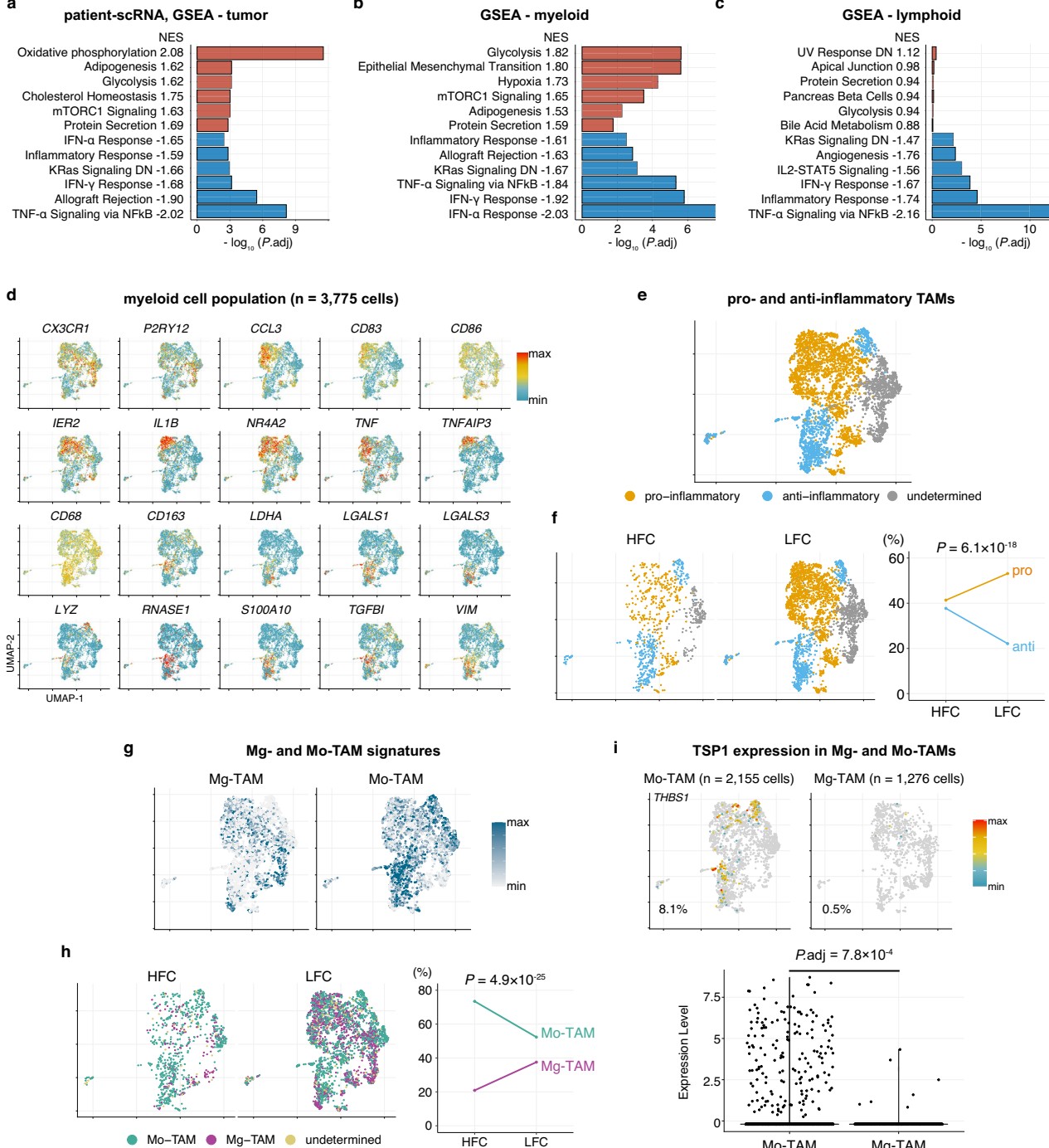

**Fig. 1 | Distinct immune-related gene expression programs and tumor-associated macrophage compositions in functionally connected intratumoral regions of human glioblastoma. a–c** Bar plots summarizing the results of gene set enrichment analyses (GSEA) with MSigDB Hallmark collection comparing HFC vs. LFC within tumor cells (**a**), myeloid cells (**b**), and lymphoid cells **c**. Statistical values are shown in each figure as normalized enrichment scores (NES) and adjusted *p* values. Positive and negative NES values indicate upregulation (shown in red) and downregulation (shown in blue) in HFC compared with those from LFC regions. The top six upregulated and downregulated signatures are presented. Complete results of the GSEA are provided in Supplementary Data 1. **d** Feature plots showing the expression patterns of representative marker genes used for the cell annotations. UMAP plots showing the relative compositions of pro-inflammatory, anti-inflammatory, or undetermined subpopulation in the entire myeloid cell population (**e**), and within those isolated from intratumoral regions with HFC and LFC **f**. The percentages of each subpopulation within each region are shown on the right. **g** Feature plots highlighting the distributions of the Mg-TAM and Mo-TAM signature scores. **h** UMAP plots showing the relative compositions of Mg-TAMs and Mo-TAMs within those isolated from intratumoral regions with HFC and LFC (left), and their percentages within each region are shown in the line plot (right). **i** Feature plots and violin plots showing the distributions of *TSP1* gene expression in Mg-TAMs and Mo-TAMs. *p* values were calculated using a two-sided Fisher's exact test **f**, **h**. For details on statistical tests used in **a–c** and **i**, see "Methods". Source data are provided as a Source Data file.

comparison, *TSP1*-positive cells were more frequently found in HFC regions than in LFC regions (13.3% vs. 2.5%), in anti-inflammatory cells than in pro-inflammatory cells (11.8% vs. 3.7%), and in Mo-TAMs than in Mg-TAMs (8.1% vs. 0.5%) (Fig. 1i and Supplementary Fig. 5b–e). These findings suggest that myeloid cells in HFC regions, which are enriched with anti-inflammatory cells and Mo-TAMs, tend to express the synaptogenic factor *TSP1* more frequently, indicating their contribution to glioma-neuronal circuit remodeling.

Importantly, during the presurgical clinical imaging tests used for identifying HFC and LFC regions, the HFC and LFC voxels were equally identified within both enhancing intratumoral regions as well as FLAIR hyperintense regions, and the tissues were collected accordingly[14]. Therefore, any differences observed in TAMs are unlikely to be owing to the sampling bias. Taken together, alongside the recent discovery that tumor-intrinsic neuronal activity drives glioblastoma proliferation, the significant immunosuppression within HFC intratumoral regions could be attributed to neuronal activity-mediated modulation of TAMs and their inflammatory states, where immunosuppressive Mo-TAMs are significantly enriched.

### Inverse spatial relationship between synaptic programs and pro-inflammatory responses

Given the finding that tumor-intrinsic functional connectivity in patients with glioblastoma is heterogeneous and associated with regional immunosuppression, we aimed to investigate its spatial significance in situ. We analyzed multiple spatially-resolved transcriptomic RNA sequencing datasets, all acquired using the 10x Genomics Visium platform. In human glioblastoma data (*n* = 6)[34,35], we delineated tumor outlines based on the distribution of the copy number alteration (CNA) index, a sum of imputed chromosome-level CNAs[36], as well as the morphology observed in H&E stained images (Fig. 2a, b and Supplementary Fig. 6). Within the estimated tumor infiltration areas with mixed CNA indices, we recurrently observed inverse correlations, to varying extents, between neuro-synaptic gene signature scores (represented by the gene set *Postsynaptic Neurotransmitter Receptor Activity* [GO:MF]) and pro-inflammatory signature scores (represented by the gene sets *TNFα-via-NFκB Signaling Pathway*, *IFNγ Response*, and *Inflammatory Response* [all from Hallmark]) (Fig. 2c and Supplementary Fig. 6).

Next, we looked into preclinical models to assess whether the observed trends were recapitulated in mouse syngeneic glioblastoma models. To this end, we analyzed Visium data from mouse brains with SB28 (*n* = 2)[37–39] and GL261 (*n* = 1, GSE245263)[40] tumors (Fig. 2d and Supplementary Fig. 7a–c, i). We estimated tumor infiltration areas based on the distribution patterns of *GBM-MES* and *Neuronal Systems* (Reactome) gene enrichment scores (Fig. 2e–h and Supplementary Fig. 7d–g, j–m). In the regions with tumor infiltration of each subject, we consistently observed a strong inverse association between synaptic activity-related gene signatures and pro-inflammatory signatures across all three individual samples (Fig. 2i and Supplementary Fig. 7h, n). In addition, deconvolution analysis using xCell validated that the neuron fraction was negatively associated with the inflammatory pathways (Supplementary Fig. 8). Together, the spatial transcriptomic analyses of both human and mouse data revealed inverse correlations between neurosynaptic activity-related and immune response-related gene expression programs within the tumor infiltration areas of glioblastoma.

### Reduced excitability in mouse glioblastoma following TSP1 knockout

Recent evidence identified TSP1 as a driver of glioma-neuron interactions within HFC regions of glioblastoma[8]. This synaptogenic factor is expressed by astrocytes in intratumoral regions without neuronal activity (LFC regions), whereas malignant tumor cells are its primary source in HFC regions. TSP1 is multifunctional, suggesting potential

roles in synaptogenesis, neuronal development[41,42], and immunomodulation in both healthy and diseased states[43–46]. Therefore, we hypothesized a potential causal link between glioblastoma synaptic enrichment and co-occurring immunosuppression, mediated by the production of synaptogenesis-associated paracrine factors.

To experimentally interrogate neuronal activity-associated immunosuppression in glioblastoma, we aimed to establish a syngeneic model recapitulating HFC and LFC glioblastoma. By screening three publicly available murine RNA-seq datasets (in vitro SB28 and GL261 syngeneic glioma cell lines, and bulk normal mouse brain), we found that endogenous gene expression of TSP1 (*Thbs1*) was significantly higher in SB28 compared to the other two datasets (vs. GL261: $log_2$fold change [FC] = 4.50, adjusted $p = 7.4 \times 10^{-60}$; vs. normal mouse brain, $log_2$FC = 9.54, adjusted $p < 1 \times 10^{-300}$) (Fig. 3a). Additionally, while SB28 tumor cells do not require neuronal trophic factors for proliferation, they exhibit significantly increased proliferation when co-cultured with mouse cortical neurons (mCN) compared to SB28 monoculture (mean proliferation index: 0.37 [SB28 alone] vs. 0.57 [SB28 + mCN]; $p = 1.2 \times 10^{-5}$) (Supplementary Fig. 9). Therefore, we hypothesized that SB28 tumors could be transcriptomically HFC-like due to high TSP1 expression, and that downregulating TSP1 in SB28 could redirect their characteristics toward an LFC-like state. We generated SB28-TSP1-knockout (KO) clones using CRISPR-Cas9 (Supplementary Fig. 10a), followed by single-cell cloning. The KO status was confirmed at the genomic DNA, mRNA, and protein levels (Supplementary Fig. 10b–d). A clone that underwent nucleofection without sgRNAs was used as the wildtype control (WT, "Cas9 only").

Next, we characterized the impact of TSP1 on synaptic puncta expression in co-cultures of SB28 glioblastoma cells and mouse cortical neurons in vitro, using immunofluorescence staining and confocal microscopy (Fig. 3b and Supplementary Fig. 11a, b). In these samples, MAP2 marks neurons, synapsin-1 marks presynaptic puncta, Homer-1 marks postsynaptic puncta, and SB28 glioblastoma cells are inherently labeled by GFP. While neurite structure was comparably maintained in both conditions (Supplementary Fig. 11c), the number of co-localized pre- and postsynaptic puncta was significantly reduced in SB28-TSP1-KO–neuron co-cultures compared to SB28-TSP1-WT–neuron co-cultures, indicating a role for SB28-TSP1-WT glioma cells in promoting synaptogenesis (mean: 1.23 [WT] vs. 0.56 [KO] per 10 μm neurite; $p = 0.004$) (Fig. 3b).

Moreover, we performed calcium imaging to assess the spontaneous activity and network dynamics of neurons in co-culture with SB28-TSP1-WT and KO cells[47]. After 24 h, neurons co-cultured with SB28-TSP1-WT tumor cells exhibited a modest but consistent trend toward an increase in activity-related parameters, including increases in the total number of calcium events per neuron, as well as amplitude and kinetics (rise and fall times) of calcium currents, compared to the baseline neuron only condition (Fig. 3c, d, Supplementary Fig. 12 and Supplementary Movies 1–2). In contrast, co-culture with SB28-TSP1-KO cells tended to reduce these neuronal activity-related parameters (Fig. 3c, d and Supplementary Movie 3). Although these trends were not statistically significant, they suggest that TSP1-expressing SB28 glioblastoma cells influence neuronal activity[14].

We further characterized the SB28-TSP1-WT and KO tumors in vivo as syngeneic orthotopic models and their interactions with surrounding non-tumor cells, including neurons. Quantitative analysis of immunofluorescence staining confirmed a significant downregulation of TSP1 protein expression in the KO tumors ($p = 0.004$) (Supplementary Fig. 13). Next, we performed bulk RNA-seq on the resected tumor tissues (Supplementary Fig. 14). Importantly, DGE analyses followed by GSEA revealed that TSP1-WT tumors were significantly enriched for genes associated with synapse and circuit assembly compared to their TSP1-KO counterparts, as exemplified by

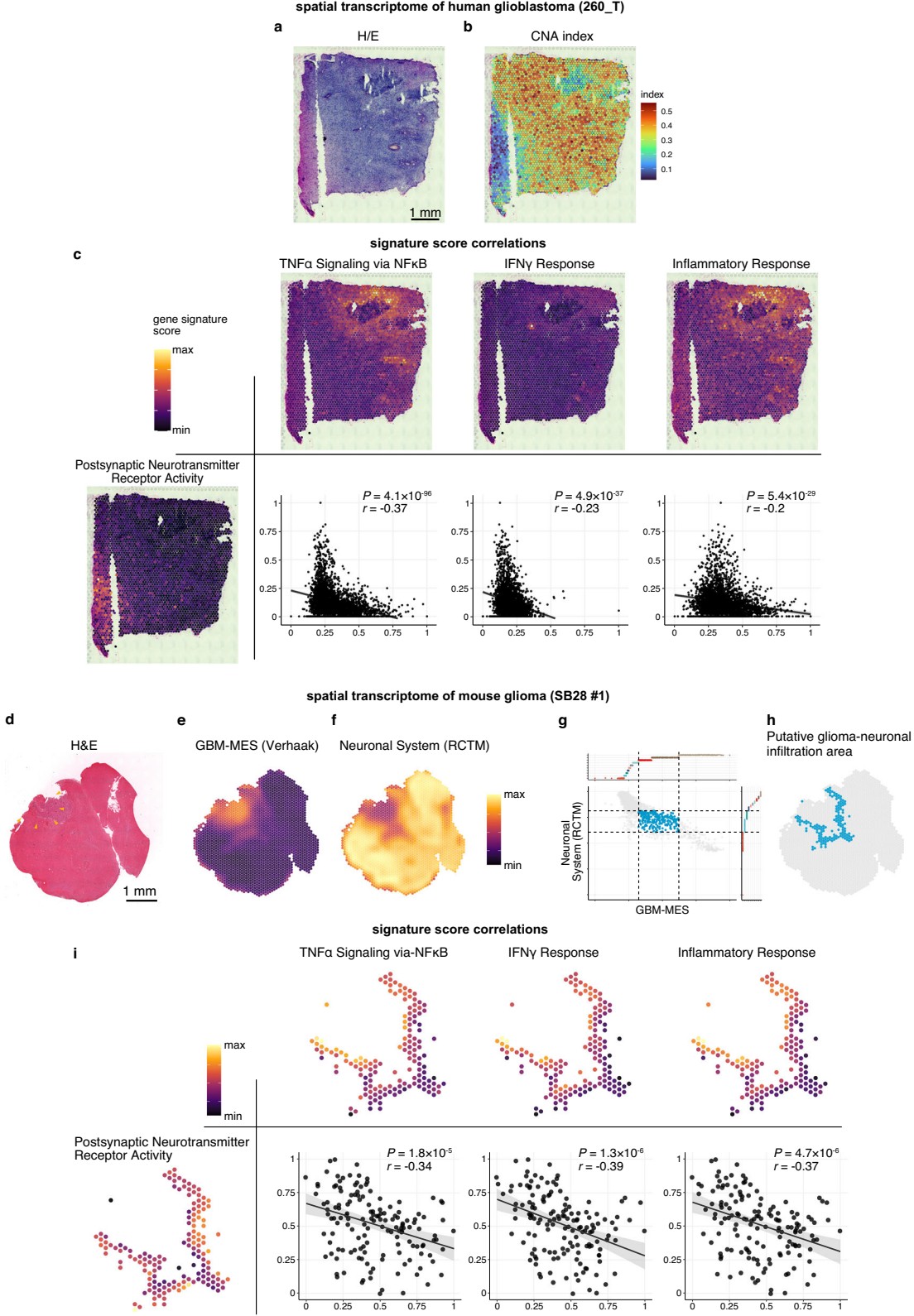

**spatial transcriptome of human glioblastoma (260_T)**

a — H/E
b — CNA index

c — signature score correlations

gene signature score

TNFα Signaling via NFκB — IFNγ Response — Inflammatory Response

Postsynaptic Neurotransmitter Receptor Activity

$P = 4.1×10^{-96}$, $r = -0.37$
$P = 4.9×10^{-37}$, $r = -0.23$
$P = 5.4×10^{-29}$, $r = -0.2$

**spatial transcriptome of mouse glioma (SB28 #1)**

d — H&E
e — GBM-MES (Verhaak)
f — Neuronal System (RCTM)
g — Neuronal System (RCTM) / GBM-MES
h — Putative glioma-neuronal infiltration area

i — signature score correlations

TNFα Signaling via-NFκB — IFNγ Response — Inflammatory Response

Postsynaptic Neurotransmitter Receptor Activity

$P = 1.8×10^{-5}$, $r = -0.34$
$P = 1.3×10^{-6}$, $r = -0.39$
$P = 4.7×10^{-6}$, $r = -0.37$

the gene sets *Neuronal System* (Reactome) and *Synaptic Transmission Glutamatergic* (GO:BP) (Fig. 3e and Supplementary Fig. 14c, d). Indeed, among the ~6300 gene sets tested, the top gene sets enriched in WT tumors were predominantly associated with neural activity and synapses, underscoring the importance of TSP1 in neuro-synaptic functions.

To further corroborate these findings, we conducted whole-cell patch clamp electrophysiology on ex vivo mouse brain slices. We recorded spontaneous excitatory postsynaptic currents (EPSCs) in L5/6 pyramidal neurons near the tumor mass and compared them between SB28-TSP1-WT and KO tumors (Fig. 3f). Both groups demonstrated similar EPSC amplitudes (mean ± s.e.m. [pA], 23.0 ± 1.9

**Fig. 2 | Spatial transcriptomic analyses reveal the inverse association between neuro-synaptic activities and immune regulation. a−c** Spatial transcriptomic analysis of a representative human glioblastoma case (sample name: 260_T). **a** Histological images (H&E) of the specimen. **b** Surface plots showing the distribution of the CNA index. **c** Surface plots displaying gene set enrichment signature scores for *Postsynaptic Neurotransmitter Receptor Activity* (GO:MF), *TNFα-Signaling* via *NFκB, IFNγ Response,* and *Inflammatory Response* (all from Hallmark) within the entire specimen. Scatter plots display the correlations between the scores of *Post-synaptic Neurotransmitter Receptor Activity* and the other pathways. **d−i** Spatial transcriptomic analysis of a representative murine glioblastoma pre-clinical model (SB28 #1). **d** Histological images (H&E). Surface plots show the

distribution of the gene set enrichment scores for *Verhaak Glioblastoma Mesenchymal* ("GBM-MES" [C2:CGP]) (**e**) and *Neuronal System* (Reactome) **f**. **g** Scatter plot showing the relationship between *GBM-MES* and *Neuronal System* scores across the entire data set, where the spots with upper 10−30 percentiles of *GBM-MES* scores and lower 10−30 percentiles of *Neuronal Systems* (Reactome) scores are highlighted in blue. **h** Surface plot showing "putative glioma-neuronal infiltration areas" defined based on the distribution of *GBM-MES* and *Neuronal Systems* scores. **i** Surface and scatter plots equivalent to (**c**) in the SB28 tumor-bearing mouse brain. *p* values were calculated using Pearson's correlation test (two-sided) (**c**) and Spearman's correlation test (two-sided) **i**. *r* correlation coefficient. Source data are provided as a Source Data file.

[WT] vs. 20.4 ± 2.4 [KO], $p = 0.41$) (Supplementary Fig. 15a). Although the mean inter-event interval (IEI) across cells was not statistically significantly different between the two groups (mean ± s.e.m. [s], 0.29 ± 0.05 [WT] vs. 0.51 ± 0.09 [KO]; $p = 0.06$) (Supplementary Fig. 15b), per-event analysis revealed a shift in the cumulative frequency distribution toward longer IEIs in the TSP1-KO group compared to the TSP1-WT group (Kolmogorov-Smirnov test, $p < 1 \times 10^{-323}$) (Fig. 3g). This result supports the presence of more excitable local networks in cortical neurons adjacent to TSP1-WT tumors. Together, these data demonstrate that SB28-TSP1-KO tumors exhibit transcriptomic and electrophysiological characteristics of TSP1-low human glioblastoma.

## TSP1 KO reprograms the glioblastoma tumor microenvironment to alleviate immunosuppression

Notably, in the same bulk RNA-seq dataset of in vivo tumors, GSEA also revealed a significant restoration of immune-related signatures in the KO tumors, such as *interferon signaling, antigen processing and presentation* via *MHC-class-I,* and *TNF-mediated signaling pathways,* compared with their WT counterparts (Fig. 4a, b and Supplementary Fig. 14e). Representative DGEs upregulated in the KO tumors included *Cd44, Tnfrsf11a/b, Nfkbia, Tap1/2,* and numerous MHC class-I/II, interferon-related, and proteasome-related genes (Supplementary Fig. 14f).

Next, we investigated brain-infiltrating leukocytes (BILs) isolated from tumor-bearing mouse brain hemispheres using flow cytometry with 1) myeloid and 2) T-cell markers. We defined CD45+CD11b+F4/80+ cells as TAMs and characterized them by staining for CD86 and CD206 (Supplementary Fig. 16a). TAMs isolated from WT tumors predominantly exhibited a CD86-CD206+ profile, suggesting an anti-inflammatory phenotype. In contrast, cells isolated from KO tumors consistently demonstrated polarization toward a CD86+CD206− profile, indicating a pro-inflammatory phenotype (Fig. 4c and Supplementary Fig. 16b). Correspondingly, the ratio of pro-inflammatory (CD86+/CD206−) to anti-inflammatory (CD86−/CD206+) TAMs was significantly higher in those isolated from KO tumors compared with their WT counterparts (mean: 0.58 [WT] vs. 1.34 [KO]; $p = 0.01$) (Fig. 4c). Moreover, we interrogated whether TAMs isolated from TSP1-WT and KO tumors would exhibit functional differences. We isolated CD11b+ cells from BILs, confirmed that over 90% were TAMs (Supplementary Fig. 17), and assessed their suppressive effects on T-cell proliferation. In a co-culture assay with carboxyfluorescein succinimidyl ester (CFSE)-labeled T-cells derived from a healthy donor mouse, CD11b+ cells from TSP1-KO tumors showed significantly less suppressive capacity on T-cell proliferation compared with their WT counterparts (proliferating CD3+ T-cells: mean: 7.5% [WT] vs. 17.7% [KO], $p = 0.03$; proliferating CD8+ T-cells: 11.3% vs. 26.8%, $p = 0.03$) (Fig. 4d and Supplementary Fig. 18). These data indicate that eliminating tumor-derived TSP1 significantly alleviates the immunosuppressive state of the tumor microenvironment.

We also performed co-culture experiments with bone marrow-derived macrophages (BMDMs) and SB28-TSP1-WT or KO tumor cells to test whether the observed differences in TAM phenotypes are

primarily attributable to tumor-secreted TSP1. Interestingly, no significant differences were observed between the effects of TSP1-WT and KO cells on the transcript levels of the tested pro-inflammatory and anti-inflammatory markers in BMDMs (Supplementary Fig. 19). The discrepancies between the in vitro and in vivo effects of SB28-TSP1-WT and KO tumor cells may suggest complex interactions within the tumor microenvironment, involving multiple cell types beyond tumor and immune cells.

Furthermore, we characterized CD45+CD3+ tumor-infiltrating T-cell populations using flow cytometry. As SB28 tumors are intrinsically characterized by sparse T-cell infiltration[38,39], we were able to recover only a limited number of T-cells, which may have impeded a comprehensive examination. Nevertheless, the analysis revealed a marked disparity in the cellular composition between the WT and KO tumors. The percentage of CD3+ T-cells within the CD45+ cells was slightly higher in KO tumors (mean: 4.3% [WT] vs. 7.1% [KO]; $p = 0.13$) (Supplementary Fig. 20a−c). Notably, the percentage of CD8+ T-cells within the CD3+ T-cell population was significantly elevated in KO tumors compared to their WT counterparts (mean: 21.4% [WT] vs. 71.5% [KO]; $p = 0.01$) (Fig. 4e). In both groups, over 95% of the CD8+ T-cell populations displayed an effector or memory phenotype, and no significant differences were observed in the expression of key activation or exhaustion markers in CD8+ T-cells between the two groups (Supplementary Fig. 20d, e). However, there was a discernible trend: PD-1 expression appeared slightly elevated in KO tumors, while TIM-3 expression was modestly higher in WT samples (Supplementary Fig. 20e).

Taken together, these findings demonstrate that TSP1 knockout in tumor cells not only suppresses glioma-associated hyperexcitability but also modulates immune responses by increasing the infiltration and functional capacity of pro-inflammatory TAMs and enhancing the abundance of CD8+ T-cells within the glioblastoma tumor microenvironment.

## AMPAR inhibition as a strategy to mitigate immunosuppression

Analyses of human clinical data and subsequent preclinical investigations, including gene perturbation studies, underscore the critical role of tumor-derived TSP1 in glioma-associated hyperexcitability and the accompanying regional immunosuppression. Furthermore, analysis of bulk RNA-seq data comparing in vivo TSP1-WT and KO tumors revealed a significant upregulation of glutamatergic signaling pathways in the WT tumors, as well as significantly higher AMPA receptor (AMPAR) gene expression scores[8] compared with their KO counterparts (Fig. 3e and Supplementary Fig. 21a). Although both glutamatergic and GABAergic signals were upregulated in the WT tumors, the shifts in glutamatergic signaling pathways were more prominent than those in GABAergic pathways (Fig. 3e and Supplementary Fig. 21b−e). Notably, TSP1 is described as a major driver of excitatory signaling by promoting glutamate release from TSP1-induced synapses[48]. Moreover, as discussed earlier, the absence of notable differences in BMDM reprogramming effects between SB28-TSP1-WT and KO cells in vitro (Supplementary Fig. 19) led us to hypothesize that excitatory neuronal

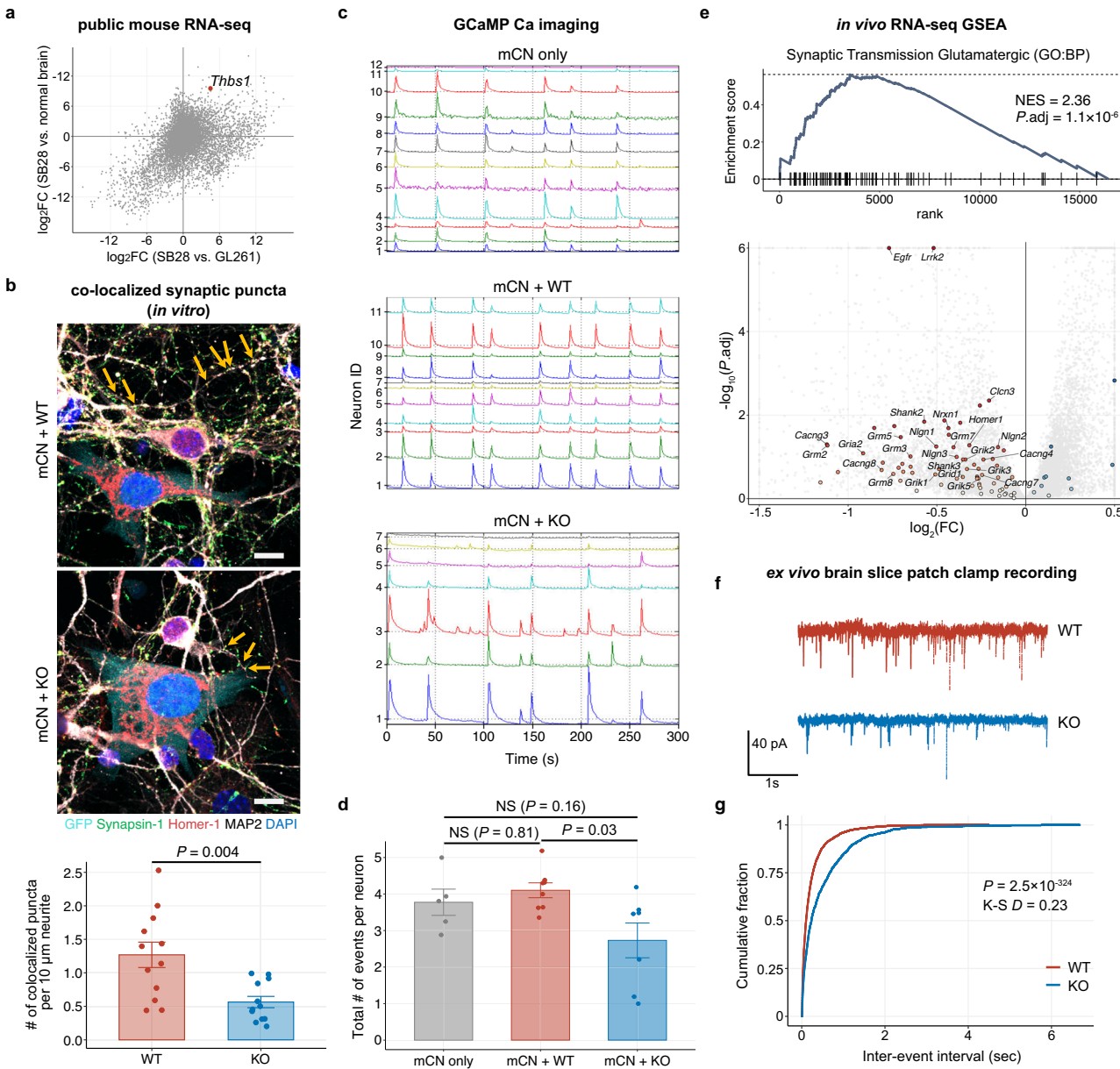

**Fig. 3 | TSP1-KO reduces glioma-associated neuronal hyperexcitability.**
**a** Scatter plot illustrating differential gene expression among three RNA sequencing datasets: SB28 (GSE127075), GL261 (GSE94239), and normal murine brain (E-MTAB-6081). *Thbs1* is highlighted in red. **b** Representative confocal images of neonatal mouse cortical neurons (mCN) co-cultured with SB28-TSP1-WT or KO cells for 24 h, showing synaptic puncta colocalization (yellow arrows). Green, synapsin-1 (presynaptic); red, Homer-1 (post-synaptic); white, MAP2 (neurons); cyan, GFP (SB28 cells); blue, DAPI. Scale bar, 10 μm. The bar plot quantifies colocalized pre- and post-synaptic puncta on neurites from 12 fields (135 μm × 135 μm) per group across 3 experiments (mean: 1.23 [WT] vs. 0.56 [KO] per 10 μm neurite; *p* = 0.004).
**c** Representative GCaMP calcium imaging traces of neonatal mouse cortical neurons alone (mCN only [top]) or co-cultured with SB28-TSP1-WT (middle) or KO (bottom) cells. **d** Bar plots comparing calcium imaging data across the three conditions, quantifying synchronized neuronal activity as calcium events per neuron (regions of interest [ROIs]): *n* = 5 for mCN only, *n* = 8 for mCN + WT, *n* = 7 for

mCN + KO). **e** Enrichment plots and volcano plots summarizing GSEA with the gene set *Synaptic Transmission Glutamatergic* (GO:BP). Positive normalized enrichment scores (NES) indicate upregulation in WT tumors vs. KO. In volcano plots, gene set members are highlighted, leading-edge genes are labeled, and genes exceeding log$_2$FC or adjusted *p* value thresholds are shown at the edges. **f** Representative traces of spontaneous excitatory postsynaptic currents (EPSCs) from pyramidal neurons identified in mouse cortical layers 5 and 6 (L5/6) near GFP-positive TSP1-WT or KO tumor lesions (400 to 800 μm in distance). **g** Cumulative frequency distributions of inter-event intervals (IEIs) from the same datasets as Supplementary Fig. 15b (WT: *n* = 2938 events; KO: *n* = 1315 events). *p* values were calculated using the two-sided Welch's unpaired *t*-test (**b**), one-way analysis of variance (ANOVA) with Tukey's post hoc test (**d**), and the two-sided Kolmogorov-Smirnov (K-S) test **g**. For details on statistical tests used in (**e**), see "Methods". Data are mean ± s.e.m. **b**, **d**. FC fold change, NS not significant. Source data are provided as a Source Data file.

synaptic activity may drive immunosuppression in the context of hyperexcitable glioblastoma. To test this hypothesis, we selected perampanel (PER), an FDA-approved, anti-epileptic drug currently used in clinical practice and under investigation in clinical trials for glioblastoma[49,50], for evaluation in an in vivo syngeneic orthotopic model.

First, we investigated the BILs isolated from PER-treated and untreated mouse brains. Flow cytometry analysis showed that CD45+CD11b+F4/80+ TAM populations in the PER-treated group were less polarized toward the anti-inflammatory phenotype and more toward the pro-inflammatory phenotype compared to the control group (the ratio of pro-inflammatory [CD86+/CD206−] to anti-

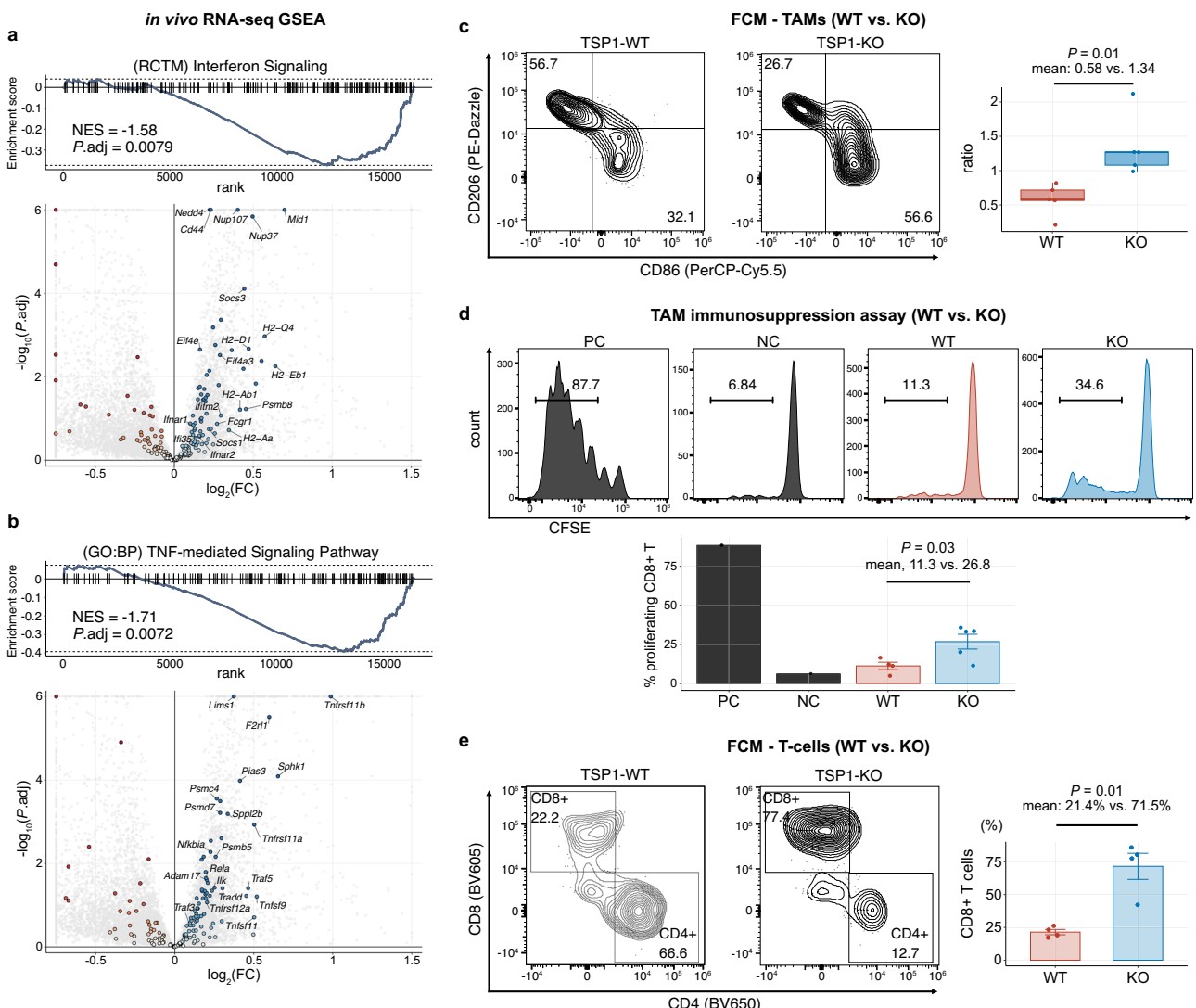

**Fig. 4 | TSP1-KO alleviates glioma immunosuppression.** Enrichment plots and volcano plots summarizing GSEA with the gene sets *Interferon Signaling* (Reactome) (**a**) and *TNF-mediated Signaling Pathway* (GO:BP) **b**. Negative normalized enrichment scores (NES) indicate downregulation in WT tumors vs. KO. In volcano plots, gene set members are highlighted, leading-edge genes are labeled, and genes exceeding log$_2$FC or adjusted *p* value thresholds are shown at the edges. **c** Flow cytometry of brain-infiltrating leukocytes (BILs) using a tumor-associated macrophage (TAM)-related marker panel in mice with SB28-TSP1-WT or KO tumors. Representative contour plots show CD86+CD206− "pro-inflammatory" and CD86−CD206+ "anti-inflammatory" populations within CD45+CD11b+F4/80+ TAMs. Box plot shows the ratio of CD86+/CD206− to CD86−/CD206+ populations (*n* = 5 mice per group; *p* = 0.01; mean, 0.58 [WT] vs. 1.34 [KO]). Values in contour plots are percentages of gated populations. The box plot shows the median (center line), interquartile range (box limits), and minimum and maximum values

(whiskers). **d** Histogram and bar plot summarizing the suppressive effect of TAMs on CD8+ T-cell proliferation. CD11b+ BILs were isolated from mice with SB28-TSP1-WT (*n* = 5) or KO (*n* = 4) tumors and co-cultured for 72 h with carboxyfluorescein succinimidyl ester (CFSE)-labeled T-cells isolated from a non-tumor-bearing mouse. The histograms show the peak distributions of CFSE signals in positive (PC) and negative controls (NC) and representative samples from WT and KO groups. The bar plot displays the percentages of proliferating T-cells. **e** Flow cytometry of BILs from mice with SB28-TSP1-WT or KO tumors, using a T-cell-related marker panel. Representative contour plots distinguish CD8+ and CD4+ populations within CD45+CD3+ T-cells. The bar plot shows the percentages of CD8+ T-cells identified in each sample (*n* = 4 mice per group; *p* = 0.01; mean: 21.4% [WT] vs. 71.5% [KO]). *p* values were calculated using the two-sided Welch's unpaired *t*-test (**c**–**e**). For details on statistical tests used in (**a**) and (**b**), see "Methods". In bar plots, data are mean ± s.e.m. **d**, **e**. Source data are provided as a Source Data file.

inflammatory [CD86−/CD206+]: mean: 0.83 [Ctrl] vs. 1.63 [PER]; *p* = 0.001) (Fig. 5a and Supplementary Fig. 22). Moreover, in a co-culture assay using CFSE dye-labeled T-cells, TAMs isolated from PER-treated mice displayed a trend toward reduced suppressive capacity on T-cell proliferation compared to the control group, although not statistically significant (proliferating CD3+ cells: mean: 7.7% vs. 14.8%, *p* = 0.08; proliferating CD8+ cells: 12.1% vs. 23.2%, *p* = 0.06) (Fig. 5b and Supplementary Fig. 23). These results were similar to the observed differences between TSP1-WT and KO tumors. Regarding T-cell profiles, while we observed a trend toward increased CD8+ T-cells in PER-treated mice, the increase was not statistically significant and was less

pronounced than that observed in TSP1-KO compared to TSP1-WT tumors (Supplementary Fig. 24a). Similarly, when evaluating activation and exhaustion markers, no significant changes were observed (Supplementary Fig. 24b, c). As for survival, consistent with previous studies testing glutamate modulation[51], treatment with PER significantly prolonged the overall survival of C57BL/6J mice bearing SB28-TSP1-WT tumors, although it was not curative (median survival: 23 days [vehicle] vs. 27.5 days [PER], Log-rank test *p* = 0.01) (Fig. 5c). Interestingly, when we treated SB28-TSP1-KO tumor-bearing mice with PER or vehicle control, no significant differences in survival durations or surface marker expression profiles on BILs were observed (Supplementary

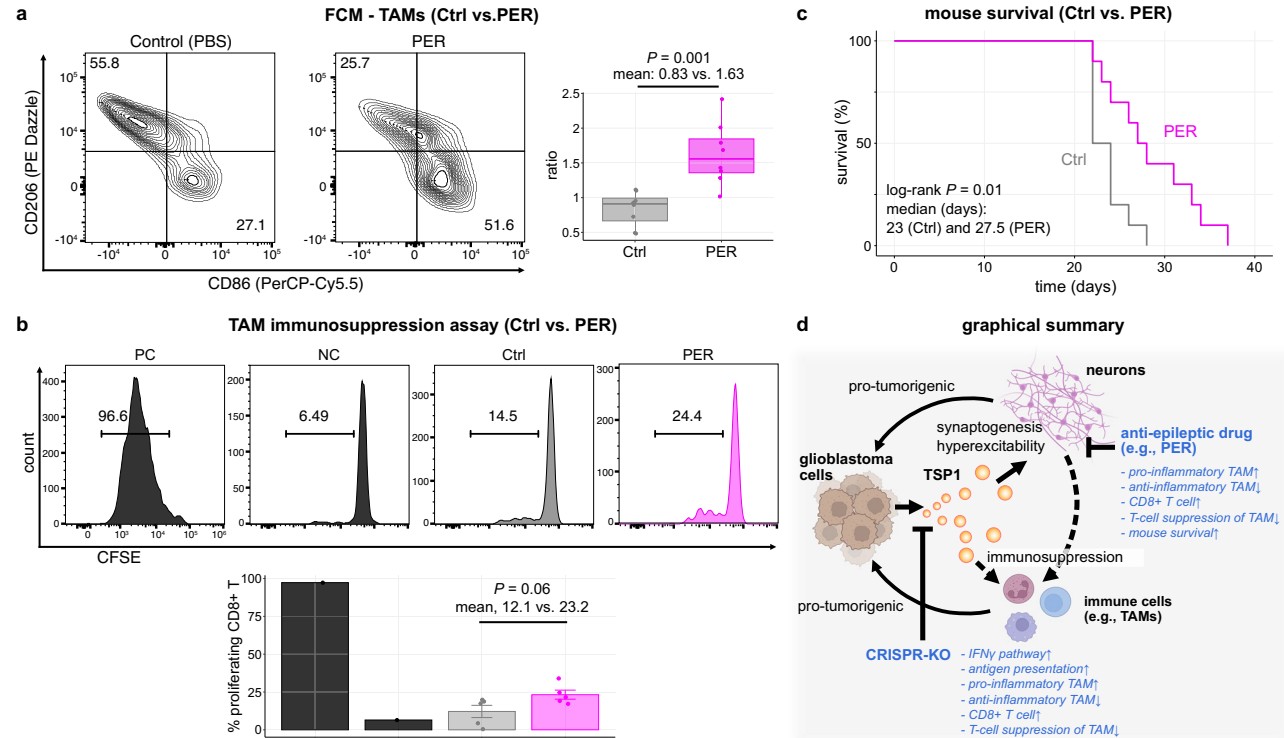

**Fig. 5 | TSP1-mediated immunosuppression and therapeutic implications of targeting glutamatergic excitatory signals. a** Flow cytometry of brain-infiltrating leukocytes (BILs) using a tumor-associated macrophage (TAM)-related marker panel in perampanel (PER)-treated and untreated mice with SB28-TSP1-WT tumors. The data were analyzed and displayed as in Fig. 4c (*n* = 8 mice per group; *p* = 0.001; mean: 0.83 [Ctrl] vs. 1.63 [PER]). The box plot shows the median (center line), interquartile range (box limits), and minimum and maximum values (whiskers). **b** Histogram and bar plot summarizing the suppressive effect of TAMs on CD8+ T-cell proliferation. CD11b+ BILs were isolated from PER-treated and untreated mice with SB28-TSP1-WT tumors (*n* = 5 mice per group). The data were analyzed as

in Fig. 4d. Data are mean ± s.e.m. **c** Kaplan–Meier survival curves of C57BL/6J mice orthotopically inoculated with SB28-TSP1-WT cells (10,000 cells/1 μL/mouse) and treated with PER (0.75 mg/kg), or vehicle control (Ctrl) via oral gavage, starting the day after tumor inoculation (*n* = 10 mice per group). **d** Schematic representation of interactions among glioblastoma cells, neurons, and immune cells, highlighting the key role of TSP1 in crosstalk and summarizing key findings from this study. *p* values were calculated using the two-sided Welch's unpaired *t*-test (**a**, **b**) and the Log-rank test **c**. PC positive control, NC negative control, NS not significant. Source data are provided as a Source Data file. The figure was created in BioRender. Nejo (2025) https://BioRender.com/z1dmf4k **d**.

Figs. 25 and 26). These findings align with the notion that glutamatergic excitatory signaling is a significant outcome of TSP1-induced synapses[48]. As such, it is plausible that the mechanisms of action for TSP1 elimination and PER treatment may overlap and converge to plateau. These results highlight the potential of neuronal activity-oriented therapeutic interventions, such as PER, to reshape the tumor microenvironment of hyperexcitable glioblastoma toward a less immunosuppressive state.

Lastly, we explored the potential additive or synergistic effects of PER in combination with immunotherapeutic strategies. As described earlier, both TSP1 elimination in tumor cells and PER treatment modulated the glioblastoma tumor microenvironment toward a more pro-inflammatory phenotype. Nonetheless, neither intervention proved curative in our preclinical investigations. This could be attributed to the limited abundance of CD3+/CD8+ T-cells and the elevated expression of exhaustion markers, even after the interventions described above. These observations prompted us to investigate combination immunotherapy involving adoptive T-cell therapy.

Specifically, we tested anti-EGFRvIII CAR T-cell therapy in combination with anti-PD-1 antibody, with or without PER, in mice bearing EGFRvIII-expressing SB28 (TSP1-WT) tumors (Supplementary Fig. 27a, b)[52–54]. After randomization into two groups, the mice were treated with an anti-PD-1 antibody (twice per week, six doses in total) and either PER or a control, starting on day 7. Additionally, all mice underwent systemic lymphodepletion on day 11 post-tumor

inoculation, followed by intravenous infusion of anti-EGFRvIII-CAR T-cells (1 × 10⁶ cells per mouse) on day 12. Regarding the survival benefit, the difference did not reach statistical significance: the median survival was 42 days in the PER-treated group compared to 39 days in the control group (Log-rank *p* = 0.34) (Supplementary Fig. 27c). However, intriguingly, complete tumor eradication was observed in 4 out of 9 (44%) PER-treated mice, compared to 2 out of 9 (22%) in the control group (Fisher's exact test, *p* = 0.62) (Supplementary Fig. 27d). On day 50 post-inoculation, after repeatedly observing the absence of tumor BLI signals, we euthanized all surviving mice, isolated BILs, and analyzed them by flow cytometry (Supplementary Fig. 27e). Among the five samples available (two from the control and three from the PER-treated group), CD45.1+ infused CAR T-cells were detected in all PER-treated mice (range, 0.35–2.75% within CD3+ T-cells) but were absent in one of the two control group mice (Supplementary Fig. 27f). Over 80% of the persistent CAR T-cells were CD8+ T-cells, and nearly all cells expressed activation and exhaustion markers (Supplementary Fig. 27g, h). Notably, a memory phenotype (defined by CD62L +CD44+), which is associated with long-term tumor remission after CAR T-cell therapy[55,56], was observed exclusively in the PER-treated group, but not in the control group (Supplementary Fig. 27i). This may suggest a potential role of neuronal activity-targeted intervention in enhancing CAR T-cell persistence. Although these descriptive data indicate a modest additive effect of PER, they underscore the need for further investigation into this therapeutic approach.

## Discussion

Advancements in cancer neuroscience research have revealed unique features and treatment resistance mechanisms specific to brain tumors[2–9,14]. Furthermore, the contributions of the immune axis to neurofibromatosis type 1 (NF1) low-grade glioma growth have also been investigated[57]. More recently, Drexler et al. epigenetically defined the neural signature of glioblastoma and demonstrated its anticorrelation with the immune component[58]. Building on this knowledge, our interdisciplinary investigation integrates cancer neuroscience, cancer immunology, and neuroimmunology to examine how glioblastoma remodels neuronal circuits to evade immune surveillance.

In this study, we uncover a previously unrecognized process in which glioma-neuronal interactions contribute to regional immunosuppression within the cortex remodeled by glioblastoma infiltration. Our findings bridge the gap between the well-established concepts of cancer neuroscience[2–14] and glioma-associated immunosuppression[21,22,27]. Expanding on the foundational knowledge established by Krishna et al.[14] regarding TSP1's role in glioma-neuronal circuit interactions, we mechanistically demonstrate that TSP1-expressing glioblastoma regions with HFC exhibit significant suppression of key immune response pathways, accompanied by an increase in immunosuppressive TAMs (Fig. 5d). Spatial transcriptomic analysis in human cases and preclinical glioblastoma models reveals an inverse relationship between neuronal activity-related and inflammatory response signatures. Further, our SB28-TSP1-WT/KO model determines the role of TSP1 in modulating both the cellular composition and immune functionality of the tumor microenvironment. These findings confirm that glioma-neuronal circuit remodeling is strongly linked with regional immunosuppression, corroborating recent findings from other studies[57,58].

Beyond these mechanistic insights, our study advances experimental models for investigating glioma-neuronal-immune interactions. By leveraging the syngeneic SB28 mouse glioma model and integrating in vitro, in vivo, and ex vivo systems, we provide a framework to dissect these interactions under controlled conditions. Moreover, our findings have direct therapeutic implications, as we show that inhibiting glutamatergic signaling can reverse immunosuppression, potentially enhancing the efficacy of immunotherapies. While previous studies have explored the role of perampanel (PER) and other glutamate receptor antagonists in glioma-neuron interactions[4,8], our study is among the first to link these treatments with immune modulation[51]. Although our preclinical results did not reach statistical significance in combination therapy settings, they lay the groundwork for future investigations into potential synergy between anti-epileptic drugs and immunotherapy. Given the parallels between our findings and clinical observations[49,50], targeting glioma-neuronal crosstalk may represent a promising avenue for therapeutic intervention.

Despite the progress made in our study, several important questions remain to be answered. Firstly, the involvement of other cell types, such as astrocytes, in regulating inflammatory responses requires further investigation, particularly in light of recent studies[59,60]. Secondly, the specific neuronal activity-related molecules responsible for immunomodulation have yet to be identified. Our in vivo observations reveal favorable effects of PER treatment on mouse survival and TAM polarization toward a less anti-inflammatory state, indicating that the impact of TSP1 on the immune tumor microenvironment in glioblastoma is likely mediated in a neuronal activity-dependent manner. Additionally, the exclusive use of SB28 model represents an essential limitation of this study, and future investigations employing additional syngeneic models will help strengthen and expand our findings. Conversely, synaptic input to glioblastoma cells might also alter their transcriptomic profile, potentially affecting the production and release of TSP1. Further investigations are necessary to explore these hypotheses and to elucidate the molecular mechanisms underlying the bidirectional communication between glioma cells and neurons, which regulate the recruitment and phenotype of immune cells, including TAMs. Thirdly, as indicated in this study testing the combination of CAR-T, ICB, and PER, there is potential for therapeutic interventions targeting neuronal activity to enhance the efficacy of immune cell therapies. While our data demonstrate promising reprogramming of the immunosuppressive tumor microenvironment, more research is warranted to advance cancer immunotherapy against glioblastoma. Addressing these questions will enhance our understanding of the mechanisms underlying glioma-neuron-immune crosstalk and open new avenues for cancer immunotherapy combined with strategies targeting glioblastoma's neuronal activity.

## Methods

### Study approval

For all human tissue studies, written informed consent was obtained from all patients, and tissue samples were used in accordance with the University of California, San Francisco (UCSF) institutional review board (IRB) for human research as previously described[14]. All the experiments and analyses using clinical samples were conducted according to the Declaration of Helsinki. All the mouse studies were performed following the protocol (protocol number AN185402-02) approved by the Institutional Animal Care and Use Committee (IACUC) of UCSF.

### Single-cell gene-expression data processing and analysis

Single-cell RNA sequencing on patient clinical samples was performed as previously reported[14]. The data has been deposited at the NCBI Gene Expression Omnibus and made publicly available under the accession code GSE223063. The resulting FASTQ files were processed using CellRanger v3.0.2 (10x Genomics) for alignment to the hg38 reference genome. The resulting filtered count matrix was further processed using R (v4.1.2) and R package Seurat (v4.0.3), including normalization and scaling using SCTransform[61], batch-effect correction using Harmony[62], and dimensional reduction with PCA and UMAP, as previously reported[14]. Based on the previously defined cell type annotations, in the present study, the following four clusters were subsetted and analyzed separately: tumor, myeloid and lymphoid cells, and astrocytes. After removing hemoglobin- and ribosome-related genes from the dataset, raw count data of each subset was normalized and scaled again. Differential gene expression analyses were performed within each cell type to compare HFC and LFC using a hurdle model tailored to scRNA-seq data, part of the MAST software package[63]. The output was sorted based on the log2FoldChange values to prepare the rank object. Preranked gene set enrichment analysis (GSEA) was performed using fgsea (1.18.0)[64] for the gene set collection Hallmark (msigdbr 7.4.1). The normalized enrichment score (NES) and adjusted $p$ values were calculated based on 1000 permutations, the program's default setting. Statistical significance was assessed using a one-sided permutation test. $p$ values were calculated, and multiple testing correction was applied using an adaptive multilevel Monte Carlo sampling scheme, as implemented in the fgseaMultilevel() function of the fgsea R package. To visualize the data in violin plots and feature plots, signature scores for *Inflammatory Response*, *Interferon-gamma Response*, and *TNF-alpha Signaling* via *NFκB* (all from the Hallmark collection) were calculated using the AddModule() function of Seurat package. All genes within each gene set were included as input features. Additionally, gene signature score analysis was also performed with pseudo-bulk approach[65]. In this method, scRNA-seq gene expression cell counts were aggregated by cell population (e.g., myeloid cells from the HFC regions of patient 1), allowing for comparisons between HFC and LFC groups ($n = 3$ pairs) for each cell type. The resulting pseudo-bulk gene expression count matrix consisted of six columns (three HFC and three LFC) for each cell type. Gene expression signature scores were then calculated for *inflammatory response*, *interferon-γ response*, and *TNFα signaling* via *NFκB* pathways, using four algorithms:

GSVA, ssGSEA[66], AUCell[67], and JASMINE[68]. Scores were subsequently z-normalized and visualized.

In-depth characterization was performed by subsetting cells annotated as myeloid cell populations ($n = 3775$ cells). Clusters were identified using shared nearest neighbor-based (SNN-based) clustering using the first 30 principal components with $k = 30$ and resolution = 0.2. A total of 6 clusters were initially identified, and then manually curated into 3 clusters (pro-inflammatory, anti-inflammatory, and undetermined) based on the expression of known marker genes[26,28]. Signature scores for Mg-TAM and Mo-TAM were calculated using the *AddModule()* function of Seurat package. The gene expression data of *P2RY12, CX3CR1, NAV3, SIGLEC8, SLC1A3* were used for Mg-TAM, and those of *TGFBI, ITGA4, IFITM2, FPR3, S100A11, KYNU* were used for Mo-TAM, respectively[26]. A cell was classified as a Mo-TAM or Mg-TAM if its corresponding signature score was both greater than 0 and higher than the alternative score. Cells with both scores below 0 were labeled as undetermined. Differential gene expression analyses, including *THBS1*, were conducted to compare different cell states or groups using a hurdle model tailored to scRNA-seq data, implemented in the MAST software package[63]. $p$ values (two-sided) were adjusted for multiple testing using the Benjamini-Hochberg method.

## Culture of tumor cell line

A C57BL/6J-background murine glioblastoma cell line, SB28[37–39], and the derived cells (passage number 12–30) were maintained in complete RPMI [RPMI 1640 media supplemented with 10% FBS, 1% Penicillin-Streptomycin (Gibco, 15070063), 1% HEPES (Gibco, 15630080), 1% Glutamax (Gibco, 35050061), 1% MEM non-essential amino acids (Gibco, 11140076), 1% sodium pyruvate (Gibco, 11360070), and 0.1% β-mercaptoethanol (Gibco, 21985023), termed cRPMI]. The cell line was originally developed in our laboratory and is now distributed through the DSMZ-German Collection of Microorganisms and Cell Cultures (https://www.dsmz.de/collection/catalogue/details/culture/ACC-880). Additionally, murine EGFRvIII (mEGFR)-expressing SB28 cell line was established through lentiviral transduction as previously reported[53,54]. Cells were passaged when reaching subconfluent every 3–4 days using Accutase (AT104, Innovative Cell Technologies), and maintained in a humidified incubator in 5% CO2 at 37 °C. All cells were routinely confirmed to be negative for mycoplasma infection every 3–4 months using PlasmoTest mycoplasma detection kit (InvivoGen, catalog # rep-pt1). No other authentication assay was performed.

## Orthotopic mouse glioblastoma models

All mouse experiments were performed following the protocol approved by IACUC of UCSF. For orthotopic syngeneic models, SB28 tumor cells were implanted intracerebrally into 5–7 week-old female C57BL/6J mice (Jackson Laboratory, 000664) with 5–10 mice per group. Male mice were not included, as sex-based differences were not specifically investigated in this study. Exact numbers for each cohort are provided in the corresponding figure legend. The surgical procedure used in the current study has been described previously[69,70]. Briefly, animals were anesthetized with 1.5–3% isoflurane and placed in a stereotactic frame (Kopf). After disinfection with betadine and ethanol and making a midline scalp incision, the injection site was located 2 mm to the right of the bregma. A burr hole was drilled at the injection site using a 25 G needle. A Hamilton syringe loaded with tumor cells (approximately $1 \times 10^4$ cells in 1 μL sterile HBSS) and equipped with a 26 G needle was inserted into the brain at a depth of 3.5 mm from the skull and then slowly pulled back to a depth of 3.0 mm to create space for the cells. The cells were injected targeting the right striatum at a speed of 1 μL/min using an autoinjector system. After the infusion, the syringe needle was held in place for 1 min, then pulled back to a depth of 1.5 mm and held in place for another min before being withdrawn slowly to minimize backflow of the injected cell suspension. The burr hole was sealed with bone wax. Aseptic techniques were used throughout the surgical procedure. Post-operatively, animals were treated with an analgesic (meloxicam and buprenorphine) and monitored for adverse symptoms in accordance with the IACUC-approved protocol.

## Bioluminescence imaging

Tumor engraftment and progression were monitored by luminescence emission on an IVIS imaging system (Xenogen). Mice were anesthetized with isoflurane and intraperitoneally (i.p.) injected with 1.5 mg of d-luciferin (GoldBio) in a total injection volume of 100 μL. The average radiance signal was used to generate all tumor growth data.

## Spatial transcriptomics data acquisition

Spatial resolved transcriptomic data was acquired for the mouse brain tissues harboring SB28 tumors ($n = 2$ mice) using the Visium Spatial for FFPE Gene Expression Kit, Mouse Transcriptome (10x Genomics, 1000339). Tissue Optimization and Library preparation were carried out according to the manufacturer's protocol (10X Genomics, CG000408 Rev A). Briefly, formalin-fixed, paraffin-embedded (FFPE) tissue blocks were made from mouse brain tissue harboring tumors collected immediately post-euthanasia and cardiac perfusion with PBS. Tissues were placed in 4% Paraformaldehyde for 24-h fixation and replaced with 70% EtOH until ready for processing. Tissues were processed and embedded into FFPE blocks on the Sakura VIP 6 and Tissue Tek 5 embedder, respectively, at the UCSF Neurosurgery Brain Tumor Center (BTC) Biorepository. Quality control of tissue blocks was performed by extracting RNA from FFPE samples using the Qiagen RNeasy FFPE Kit (Qiagen, 73504), followed by the assessment on the Agilent 2100 Bioanalyzer using the RNA 6000 Pico Kit (Agilent, 5067-1513). The DV200 values were confirmed to be 67% and 70%, meeting the minimum requirement of no less than 50%. Then, 5 μm-thick sections were mounted onto each spatially barcoded capture area of the Visium Spatial Gene Expression slide. The mounted slide was dried by storing in a desiccator at RT overnight, and finally at 60 °C for 2 h. Deparaffinization was performed with xylene; then, the slide was immediately stained with hematoxylin and eosin and imaged at ×60 magnification at the Gladstone Institute Histology and Light Microscopy Core. The subsequent library preparation, quality control, and sequencing steps were performed at the Gladstone Institute Genomics Core. Tissue sections were de-crosslinked at 70 °C for 1 h, then hybridized overnight with the mouse transcriptome probes. Probe ligation was followed by the release of single-strand product from the tissue and binding to the Visium slide. Probe extension then added the unique molecular identifier (UMI), spatial barcode, and partial Read 1. Following the elution of samples from the Visium slide, one microliter of each sample was subjected to 25 cycles of qPCR. The optimal amplification cycles were determined using the Cq values at the exponential phase of the amplification plot, which is roughly 25% of the peak fluorescence. These values were used in the subsequent library preparation steps, where dual indexes were added to the barcoded products. Quality control of the final libraries was completed on the Agilent 2100 Bioanalyzer using the High Sensitivity DNA Kit (Agilent, 5067-4626) to determine the average library sizes, in addition to qPCR on the Applied Biosystems QuantStudio 5 Real-Time PCR System using the Roche KAPA Library Quantification Kit (Roche, KK4824) to determine the concentration of adapter-ligated libraries. Finally, libraries were pooled and sequenced on the NextSeq 500 high output 150 cycle flow cell, paired-end 28 × 50 bp with 10 bp dual indexes, resulting in greater than 44,000 paired reads per capture spot.

## Spatial transcriptomics data processing

Space Ranger v1.3.1 (10x Genomics) was used to integrate the FASTQ sequencing files and the H&E staining image files, construct initial count matrices for each unique molecular identifier (UMI) at every location in each sample using the mouse reference (refdata-gex-mm10-2020-A),

and generate output files for subsequent analyses. Downstream analysis and visualization were done using R (v4.1.2) and the R package SPATA2 (v2.0.4) (https://github.com/theMILOlab/SPATA2)[34]. Denoising of the data was performed using the *runAutoencoderDenoising()* function from the package. All subsequent analyses were performed on the denoised expression data. Gene expression signature scores were calculated using the *AddModuleScore()* function from Seurat. To perform region-specific analyses, the tumor bed was identified morpholotically as well as by referring to the distribution of gene expression signatures, such as *HM_HYPOXIA*, *CELL_CYCLE* (C5:GO:BP), and *VERHAAK_GLIO-BLASTOMA_MESENCHYMAL* ("*GBM-MES*" [C2:CGP]). Then the tumor bed region was approximately delineated with adequate margins using the *createSpatialSegmentation()* function of SPATA2. Next, gene signature scores for *GBM-MES* and *NEURONAL_SYSTEMS* (C2:CP:Reactome) were calculated across all spots in the entire dataset. Based on their distribution patterns, spots with scores in the upper 10–30 percentiles of *GBM-MES* and lower 10–30 percentiles of *Neuronal Systems* were identified. Finally, spots located outside the tumor bed regions were excluded, and the remaining spots were defined as "putative glioma-neuronal infiltration areas". For the spots identified as infiltration area, the gene expression signature scores for *TNFA_SIGNALING_VIA_NFKB*, *INTERFERON_GAMMA_RESPONSE*, *INFLAMMATORY_RESPONSE* (all from the Hallmark collection) and *POSTSYNAPTIC_NEUROTRANSMITTER_RECEPTOR_ACTIVITY* (C5:GO:MF) were recalculated using the *AddModuleScore()* function. Cell-type deconvolution analysis was performed on the transcriptome data of each spot within the glioma-neuron infiltration area ($n = 148$ spots) using xCell[71]. From the output data matrix, which contained calculated scores of 66 cell types and 3 scores for each spot, we carefully curated 22 cell types, comprising 18 immune cells and 4 central nervous systems cell types (astrocytes, endothelial cells, neurons, and pericytes). Next, Pearson's correlation scores were calculated among the 22 cell types and 6 relevant gene signature scores, as visualized in Supplementary Fig. 8.

## Analysis of publicly available spatial transcriptome data

Spatial transcriptome data of human glioblastoma specimen were obtained from SPATAData (https://github.com/theMILOlab/SPATAData)[34,35]. Downstream analysis and visualization were carried out using R (v4.1.2) and the R package SPATA2 (v2.0.4) (https://github.com/theMILOlab/SPATA2)[34,35]. Chromosomal copy number variation was assessed using the *runCNV()* function in SPATA2. The copy number alteration (CNA) index was calculated by summing the absolute deviations of inferred CNVs from 1 (neutral), providing an overall assessment of the CNA burden[36]. Six samples with relatively heterogeneous distributions of CNA index were manually curated (248_T, 259_T, 260_T, 275_T, 296_T, and 304_T). For each specimen, gene expression signature scores for *TNFA_SIGNALING_VIA_NFKB*, *IL2_STAT5_SIGNALING*, *INTERFERON_GAMMA_RESPONSE*, *INFLAMMATORY_RESPONSE* (all from the Hallmark collection) and *POSTSYNAPTIC_NEUROTRANSMITTER_RECEPTOR_ACTIVITY* (C5:GO:MF) were calculated across all spots on the specimen using the *AddModuleScore()* function.

Spatial transcriptome data for the murine glioma GL261 was obtained from the Gene Expression Omnibus (GEO) under accession number GSE245263[40]. The downloaded transcriptome and image data were converted from AnnData format into a SPATA object using the *asSPATA2()* function, and then analyzed using the R package SPATA2 (v2.0.4), following the same approach as used for the two SB28 samples described above.

## Analysis of public transcriptome data

We obtained three publicly available bulk transcriptome datasets: GSE127075 (SB28)[38] and GSE94239 (GL261) from Sequence Read Archive (SRA), and E-MTAB-6081 (C57BL/6 normal mouse brain)[72] from EMBL-EBI ArrayExpress ($n = 3$ samples each). The fastq reads were aligned to the mouse reference genome mm10 (GRCm38.p6) using STAR (v2.7.9a), with transcriptome annotation guidance from gencode.vM25.annotation.gtf. Sorting and indexing were performed using samtools (v1.14), and gene-level expression counts were estimated using stringtie (v2.0)[73]. All subsequent computational analyses were conducted using R (v4.1.2). Differential expression (DE) analyses comparing two groups were performed using the DESeq2 R package (v1.32.0)[74]. Log$_2$ fold change (FC) values from each analysis were used for data visualization.

## Gene knockout using CRISPR-Cas9 system

Gene knockout was performed using the Gene KO kit v2 (Synthego) following the manufacturer's recommended protocol. The kit contains three multi-guide sgRNAs specifically targeting regions within exon 3 (ENSMUSE00000295002) of the murine *Thbs1* gene. To prepare ribonucleoprotein (RNP) complexes, we mixed 60 pmol of multi-guide sgRNAs (20 pmol per each of the three individual gRNAs) and 20 pmol of recombinantly produced and purified Cas9-2NLS (QB3 Macrolab at QB3-Berkeley) at a molar ratio of 3:1 for sgRNA to Cas9. The mixture was incubated for 30 min at RT. For the TSP1 wildtype control (an experimental negative control), only the Cas9 protein was added without sgRNAs. Electroporation was performed using SE Cell Line 4D-Nucleofector X kit S (Lonza, V4XC-1032). SB28 parental cells at a subconfluent condition were dissociated with Accutase, washed with PBS, and 150,000 cells were added to each tube containing RNP complexes. The volume was adjusted to 20 μL with Nucleofector solution and transferred to the Nucleocuvette Vessel provided in the kit. Nucleofection was performed using the DS126 program ("MG-U87") of the 4D-Nucleofector X Unit (Lonza). Immediately after nucleofection, cells were recovered by adding 80 μL of pre-warmed growth media to each well, and then in a humidified incubator with 5% $CO_2$ at 37 °C for 30 min. The cells were collected, replated on culture dish plates, and allowed to grow. On day 4, cells were detached, dissociated, and subjected to limiting dilution and clonal expansion in a 96-well plate. The knock-out status of the *Thbs1* gene in the expanded clones was determined by amplifying genomic DNA using polymerase chain reaction (PCR), followed by Sanger sequencing as well as Western blotting, as described below. Clone 1C1 was selected as a representative knock-out clone based on induced frame-shift alterations at the genomic and transcriptomic levels, as well as robust downregulation at the protein level. This clone was used for all subsequent in vitro and in vivo experiments unless otherwise specified.

## Sanger sequencing

The DNeasy Blood & Tissue Kit (Qiagen, 69506) was used to extract genomic DNA from the culture cells, according to the manufacturer's protocols. An aliquot of genomic DNA was amplified by polymerase chain reaction (PCR) using the following oligo primers. Fwd: 5'-TAAGGATGCAGCTTCCCTCG-3'; Rev: 5'-CCGTTGGAGACCACACTGAA-3'. After the gel electrophoresis and image acquisition, the agarose gel pieces were cut out and digested using NucleoSpin® Gel and PCR Clean-Up kit (Takara, 740609), according to the manufacturer's protocols. Sequencing was performed at Quintara Biosciences, using the following sequencing primer: 5'-TTTCCATAATTGCCATTATT GTCACGAGTT-3'. Acquired Sanger sequencing data was analyzed and visualized using ApE (v2.0.61).

## Western blot

Cultured cells were lysed with ice-cold IP lysis buffer (Thermo Fisher Scientific, 87788) containing protease inhibitor cocktail (Sigma-Aldrich, 11836170001) and phosphatase inhibitor (Sigma-Aldrich, 04906845001) to prepare the total cell lysate. The protein concentration of the lysate samples was determined using a BCA assay (Thermo Fisher Scientific, 23227), and the input protein amount for

western blot analysis was adjusted accordingly. The lysate samples were denatured by mixing with Blue Loading Gel Dye and DTT (Cell Signaling Technology, 7722). Primary antibodies used in this experiment were as follows: TSP1, anti-Thrombospondin-1 clone A6.1 (Thermo Fisher Scientific, MA5-13398) at a 1:500 dilution, and GAPDH, anti-GAPDH clone 14C10 (Cell Signaling Technologies, #2118) at a 1:1000 dilution. Secondary antibody staining employed anti-mouse IgG and anti-rabbit IgG HRP-linked antibodies (Cell Signaling Technologies, #7076 and #7074, respectively) were used at 1:5000 dilution. The primary antibody staining step was performed overnight at 4 °C, followed by the secondary antibody staining at RT for 1 h. To visualize the protein size ladder, Precision Plus Protein™ WesternC™ Blotting Standards and Precision Protein™ StrepTactin-HRP Conjugate (Bio-Rad, 1610376 and 1610381) were used. Bullet Blocking One for Western Blotting (Nacalai USA, 13779-01) was used for blocking and as the antibody diluent. Western blot bands were visualized using Pierce™ ECL Western Blotting Substrate (Thermo Fisher Scientific, 32106) with the Odyssey FC imaging system (LI-COR Biotechnology), and the analysis was performed using Image Studio Software (v5.2.5, LI-COR). An unprocessed image of the whole blot is provided in the Source Data file.

## Bulk RNA-sequencing of in vitro cells

Total RNA was extracted from tumor cell pellets using RNeasy Mini Kit (Qiagen, 74106) and RNase-Free DNase Set (Qiagen, 79254) following the manufacturer's protocol. RNA integrity was evaluated using Agilent Bioanalyzer 2100 and confirmed to be RIN 9.8 or greater. The following library preparation and sequencing were performed by DNA Technologies and Expression Analysis Core Laboratory at the University of California, Davis (UC Davis) Genome Center. Strand-specific and barcode-indexed RNA-Seq libraries were generated from 300 ng total RNA each after poly-A enrichment using the mRNA-Seq Hyper Kit (Kapa Biosystems, KK8581) following the manufacturer's instructions. The fragment size distribution of the libraries was verified via microcapillary gel electrophoresis on a Bioanalyzer 2100. The libraries were quantified by fluorometry on a Qubit fluorometer (Life Technologies) and pooled in equimolar ratios. The pool was quantified by quantitative PCR with a Library Quant Kit (Kapa Biosystems, KK4824) and sequenced on an Illumina NovaSeq 6000 with paired-end 150 bp reads.

## Bulk RNA-sequencing of in vivo tumors

To perform RNA-seq with in vivo tumor tissue samples, mice were euthanized before being perfused via transcardial injection of 10 mL cold PBS. The whole brains were then harvested, and visible tumor tissues were dissected and stored in RNAlater (Invitrogen, AM7024) at 4 °C until RNA extraction (within 1 week). Total RNA was extracted using the RNeasy Mini Kit (Qiagen, 74106) and RNase-Free DNase Set (Qiagen, 79254) following the manufacturer's protocol. The tissues were homogenized using a Bioruptor standard water bath sonicator (Diagenode, UCD-200) with the following settings: power, H; duration, 5 min; cycle, 30 s/30 s. Due to the sonication treatment, the RIN scores were found to be low (4.2–6.5) when evaluated with an Agilent Bioanalyzer 2100. Because of these low RIN scores, we chose ribosomal depletion for mRNA enrichment. The subsequent library preparation and sequencing were performed by the UCSF Genomics CoLab. Starting material of 500 ng of total RNA was used according to vendor instructions with Universal plus mRNA with Nu Quant (TECAN, 0520), with noted changes to the protocol as follows. First, the Poly(A) Selection step was omitted. Second, we started with RNA fragmentation and used QIAseq FastSelect −rRNA/Globin Kit (Qiagen, 335377) to remove rRNA as follows: 0.1 μL of FastSelect rRNA and 1 μL of 1X Fragmentation Buffer was added to 10 μL of total RNA, with the addition of 10 μL of 2X Fragmentation Buffer. PCR program was used as follows: 94 °C/3 min, 75 °C/2 min, 70 °C/2 min, 65 °C/2 min, 60 °C/

2 min, 55 °C/2 min, 37 °C/5 min, 25 °C/5 min, 10 °C/hold. After the rRNA depletion step, Universal plus mRNA with Nu Quant protocol was followed, starting with the first strand cDNA synthesis step but omitting the steps of AnyDeplete and NuQuant. After final library PCR amplification of 15 cycles and bead clean-up, individual libraries were pooled equally by volume, and quantified on Fragment Analyzer (Agilent, DNF-474). The quantified library pool was diluted to 1 nM and sequenced on MiniSeq (Illumina, FC-420-1001) to check for the quality of reads. Finally, individual libraries were normalized according to MiniSeq output reads, specifically by % protein-coding genes, and were sequenced on one lane of NovaSeq6000 SP PE150 (Illumina, 20028400) at UCSF Center for Advanced Technology (CAT).

## Bulk RNA-sequencing data processing and analysis

Quality checking, trimming, and removal of barcodes, were performed using fastp (version 0.20.0) with default parameters. The fastq reads were aligned to the mouse reference genome mm10 (GRCm38.p6) using STAR (v2.7.9a), with transcriptome annotation guidance from gencode.vM25.annotation.gtf. Sorting and indexing were performed using samtools (v1.14), and gene-level expression counts were estimated using stringtie (v2.0)[73]. The mapping status was visualized using the Integrative Genomic Viewer (IGV, version 2.8.0). All subsequent computational analyses were conducted using R (v4.1.2). Differential expression (DE) analyses comparing two groups were performed using the DESeq2 R package (v1.32.0)[74]. The output was sorted based on the stat values to prepare the rank object. Preranked gene set enrichment analysis (GSEA) was performed using fgsea (1.18.0)[64] for the following gene set collections: Hallmark, C2 KEGG, C2 Reactome, and C5 GO:BP (msigdbr 7.4.1). In GSEA, statistical significance was assessed using a one-sided permutation test. p values were calculated, and multiple testing correction was applied using an adaptive multilevel Monte Carlo sampling scheme, as implemented in the fgseaMultilevel() function of the fgsea R package. The AMPAR gene expression score[8] was calculated using all four AMPAR genes (Gria1, Gria2, Gria3, and Gria4) with the GVSA R package.

## Neonatal mouse cortical neuron culture

Neonatal mouse cortical neuron cultures were prepared as described previously[14]. Neonatal (P1.5) C57BL/6J mice were euthanized by hypothermia followed by decapitation. The cerebral cortex was dissected under a microscope using aseptic techniques, with tissues from 3–6 mice pooled and processed together. To isolate cortical neurons, the cortices were minced with scalpels on an ice-cold culture dish and digested using a papain dissociation kit (Worthington Biochemical, LK003150), per the manufacturer's instructions. Briefly, papain digestion and subsequent protease inhibition were performed at 37 °C for 7 and 3 min, respectively. Cell pellets were washed with Neurobasal-A medium (Gibco, 10888022) supplemented with Deoxyribonuclease I (1 mg/mL; Worthington, LS002007). The cell suspension was filtered through a 40 μm filter, counted, and centrifuged. Neuron culture media (NCM) was prepared by mixing Neurobasal-A medium with 1× B-27 (Gibco, 17504044), 1% Glutamax (Gibco, 35050061), 1% sodium pyruvate (Gibco, 11360070), and 1% Antibiotic-Antimycotic (Gibco, 15240062). Cells were resuspended in neuron plating media (NPM), prepared by adding 4.5% fetal bovine serum (FBS) to the NCM, then plated onto poly-D-lysine and laminin-coated coverslips (NeuVitro, GG-12-1.5-laminin) at a density of $1.5 \times 10^5$ cells per well in 24-well plates (approximately $7.9 \times 10^4$ cells per cm²). After 24 h, the culture media was replaced with serum-free NCM. Cultures were maintained by replacing half the culture media with fresh NCM every 3–4 days. For co-culture experiments with SB28 glioblastoma cells for synapse imaging, SB28-TSP1-WT or KO cells (approximately $3 \times 10^3$ cells in 100 μL of NCM) were added onto the neuron cultures on coverslips at days 7–9 and maintained for 24 h.

## Immunofluorescence staining

In vitro co-cultures of neurons and tumor cells were terminated after 24 h. Following removal of the culture medium and a PBS wash, cells were fixed with ice-cold 4% paraformaldehyde at 4 °C for 30 min. Coverslips were then washed twice with ice-cold PBS and stored at 4 °C until immunofluorescence staining was performed. Staining was carried out in 24-well tissue culture plates with coverslips. Cells were blocked with a buffer containing 0.25% Triton X and 5% normal goat serum in PBS for 45 min at RT, then incubated overnight at 4 °C with primary antibodies diluted in the same blocking buffer. The primary antibodies and their dilutions were as follows: anti-MAP2 Rabbit pAb (1:500, Synaptic Systems, 188 003) for neurons; anti-Synapsin-1 Mouse mAb (1:300, Synaptic Systems, 106 011); anti-Homer1 Guinea Pig pAb (1:300, Synaptic Systems, 160 004). After two washes with PBS, cells were stained with secondary antibodies diluted at 1:250 for 1 h at RT. The secondary antibodies used were: Goat Anti-Rabbit IgG AF594 (Abcam, ab150084); Goat anti-Mouse IgG AF514 (Invitrogen, A-31555), Goat anti-Guinea Pig IgG AF647 (Invitrogen, A-21450), prepared in 0.25% Triton X and 5% normal goat serum in PBS. After washing and dehydration, the stained coverslips were mounted cell-side down on glass slides using ProLong Diamond Antifade Mountant with DAPI (Fisher Scientific, P36971). For Ki-67 proliferation index assay, the following primary and secondary antibodies were used: anti-Ki-67 (D3B5) Rabbit mAb (1:200, CST, 9129) and Goat anti-Rabbit IgG AF647 (Abcam, ab150083).

For in vivo tissue samples, mice were euthanized with $CO2$ inhalation before intracardiac perfusion with 10 mL cold PBS. The brains were harvested and fixed with 4% paraformaldehyde at 4 °C overnight, then soaked in 30% sucrose for 1–2 days to ensure adequate cryoprotection. Fixed brain tissues were subsequently embedded in Tissue-Tek® O.C.T. Compound (Sakura Finetek, 4583), and stored at 80°C. For immunofluorescence staining, serial 10-μm coronal sections were cut using a freezing microtome and fixed with 10% formalin at RT for 10 min. Sections were washed twice with 1× wash buffer (Dako, S300685-2C) and blocked with PBS containing 5% normal goat serum (Abcam, ab7481) and 1% TruStain FcX PLUS anti-mouse CD16/32 antibody (BioLegend, 156604) for 40 min at RT. The primary antibody, anti-Thrombospondin-1 Rabbit pAb (1:200, ab85762, Abcam), was applied overnight at 4 °C. After washing twice with 1X wash buffer, secondary antibody staining was performed using Goat anti-Rabbit IgG AF647 (Abcam, ab150083) at a dilution of 1:250 for 1 h at RT. Following washes and dehydration, stained samples were mounted with ProLong Diamond Antifade Mountant with DAPI (Fisher Scientific, P36971) on glass slides and covered with cover glasses.

## Tissue image acquisition and quantitative analysis

Images of in vitro cell proliferation assay samples and in vivo tissue samples were acquired using a Zeiss Axio Imager 2 microscope (×20 magnification) with TissueFAXS scanning software (TissueGnostics). Consistent exposure times and thresholds were maintained within each imaging session. Raw image data was imported into ImageJ software (version 2.9.0) for quantitative analysis. The tumor outline was manually identified in FITC channel images for GFP signal detection and used as a region of interest (ROI). Signal intensity in the Cy5 channel-images for AF647 signal detection was measured within the ROI. Background signal intensities were measured in a non-tumor area of the same tissue. TSP1 expression in the ROI was quantified as the adjusted mean fluorescence intensity, calculated as the mean signal intensities in the ROI minus the mean signal intensities of the background. For the cell proliferation assay, the proliferation index was calculated as the number of Ki67+ nuclei divided by the number of DAPI+ nuclei within GFP-expressing SB28 cells under each condition. Data were obtained from 30 randomly selected fields of view (FOVs) per group (663 μm × 663 μm each) across three independent culture experiments (10 FOVs per group per experiment).

## Confocal imaging and colocalization analysis of synapsin-1 and homer-1 staining

Five-color confocal images were captured using a ×63 oil-immersion objective on a Zeiss LSM780 confocal microscope equipped with Zeiss Zen imaging software. Z-stack images of 135 μm × 135 μm field of views (FOVs) containing GFP-positive tumor cells were obtained at resolutions of 2048 × 2048 or 4096 × 4096. Additionally, 3 × 3 tile images (405 μm × 405 μm) were taken to capture surrounding structures. The microscope settings for the synapsin-1 AF514 and homer-1 AF647 channels were kept constant across all samples throughout the experiment. The acquired images were imported into ImageJ software (version 2.9.0) for further analysis. From the Z-stack data, 2D images were constructed using the Maximum Intensity Projection algorithm. Threshold values for the channels were consistently adjusted across all samples to ensure accurate identification of events. Neurite structures were detected in a semi-automated manner based on MAP2 signals, and the neurite path length per FOV was quantified using SNT (Simple Neurite Tracer, version 4.3.0). The output of neurite path detection was exported, and the branching pattern was further analyzed with the Fiji plugin's skeletonization function. Colocalization of pre- and post-synaptic puncta, defined as the overlap or adjacency between synapsin-1-positive puncta and homer-1-positive puncta on or adjacent to the identified neurite structures, was quantified using the Multi-point tool of Fiji. The results were normalized to the number of colocalized puncta per 10 μm of neurite length for each FOV.

## In vitro calcium imaging

For calcium imaging, we dissociated postnatal pup (P2.5) cortices into single-cell suspension and plated them on poly-D-lysine and laminin-coated coverslips at a density of $2.5 \times 10^5$ cells per well in Neurobasal media containing B27, Glutamax, sodium pyruvate, and Antibiotic-Antimycotic. After 24 h of plating, primary neurons were transduced with an adeno-associated virus expressing GCaMP7f under the control of synapsin-1 promoter (Addgene # 104488)[47]. Five days post transduction, single cell suspension of SB28-TSP1-WT or KO cells ($3 \times 10^3$ cells) were added to neurons and maintained for 24 h. Following 24 h of co-culture, the coverslips were placed in Tyrode's solution (containing [in mM]: NaCl 129, KCl 5, $MgCl_2$ 2, $CaCl_2$ 2.6, glucose 30, HEPES 25; pH 7.3 adjusted with NaOH) and placed in a Warner Instruments perfusion chamber on an automated stage of an inverted epifluorescence microscope equipped with an excitation filter wheel (TE2000U; Nikon). Imaging was performed for 5 min with a time interval of 1 s using a 10× objective and time-lapse images were captured using MetaMorph software and analyzed using Fluor-oSNNAP software[75]. SB28 glioma cells were identified in the field of view based on their morphology. The raw calcium fluorescence signal for each neuron (ROI) was obtained by averaging all pixels within the ROI. The normalized fluorescence changes ($\Delta F/F_0$) were calculated by subtracting each data point of raw fluorescence with the mean of the lower 50% of previous 10-s values and dividing it by the mean of the lower 50% of previous 10-s values. Automated event detection was performed with the template-matching algorithm[76] included in the FluoroSNNAP. A time-varying correlation coefficient between fluorescence trace and calcium transient templates (from the event wave form library) was calculated. Fluorescence transients with amplitude $\Delta F/F > 0.1$ and correlation coefficient $>0.85$ were identified as events.

## Acute brain slice preparation

After $15 \pm 3$ days post tumor mass inoculation, ex vivo coronal brain slices were prepared for whole-cell patch-clamp electrophysiology recordings, as previously described[77,78]. Brain-slicing artificial cerebrospinal fluid (aCSF) equilibrated with carbogen (95% $O_2$, 5% $CO_2$) was prepared as follows (in mM): 93 N-methyl-D-glucamine (NMDG), 2.5 KCl, 1.25 $NaH_2PO_4$, 30 $NaHCO_3$, 20 HEPES, 25 glucose, 2 thiourea, 5

L-ascorbic acid, 3 Na pyruvate, 0.5 $CaCl_2$, 10 $MgSO_4$.$7H_2O$, pH 7.3-7.4, 300–310 mOsm. Mice were euthanized with $CO_2$ inhalation before being perfused via transcardial injection of 10 mL of ice-cold (6–8 °C) brain-slicing aCSF. The brain was removed quickly after decapitation, transferred to ice-cold brain-slicing aCSF, embedded in 1.8–2% agarose (BP165-25; Fisher Scientific), and sectioned at 350 μm thickness using a compresstome (VF-310-0Z, Precisionary Instruments; blade oscillation level 4, advancement speed of 1). Slices were transferred to recover in brain-slicing aCSF at 34 °C for 12 min. Then, slices were maintained at room temperature (RT, 20 °C) for a minimum of 1 h in a holding chamber filled with carbogenated recovery aCSF containing (in mM): 97 NaCl, 2.5 KCl, 1.2 $NaH_2PO_4$, 30 $NaHCO_3$, 25 glucose, 20 HEPES, 2 $CaCl_2$, 2 $MgSO_4$.$7H_2O$, 2 thiourea, 5 L-ascorbic acid, 3 Na pyruvate, pH 7.3-7.4, 300–310 mOsm, before recording. Any slices that showed evidence of physical damage in the cortex from previous tumor cell injection or demonstrated cortical deformation due to extensive tumor growth in the striatum were not used for this study.

### Whole-cell patch-clamp recordings

Ex vivo mouse brain slices were transferred to the recording chamber, superfused with carbogenated recording aCSF containing (in mM): 125 NaCl, 2.5 KCl, 1.25 $NaH_2PO_4$, 25 $NaHCO_3$, 11.1 D- (+)-glucose, 1 $MgCl_2$, 2 $CaCl_2$, pH 7.3–7.4, 295–305 mOsm at 34 °C at 2–3 mL/min. Initially, the tumor mass was visualized using infrared differential interference contrast (IR-DIC) optics and GFP fluorescent imaging with a 10× air objective lens on a BX51WI microscope (Olympus). Under a 40× water immersion objective lens, layer 5 and 6 cortical pyramidal neurons located near the tumor mass in the striatum (within 400 to 800 μm) were targeted for electrophysiology experiments. Whole-cell patch-clamp recordings of these neurons were performed using 3–5 MΩ glass recording electrodes (BF150-86-10; Sutter Instrument) filled with an internal solution containing (in mM): 120 K gluconate, 10 HEPES, 4 KCl, 4 MgATP, 0.3 NaGTP and 10 Na phosphocreatine, pH 7.25, 320 mOsm. All electrophysiological signals were amplified and digitized using Multiclamp 700B and Digidata 1550B (Molecular Devices), respectively. Spontaneous postsynaptic currents were recorded in voltage clamp configuration at −70 mV, sampled at 20 kHz, and low-pass filtered at 1 kHz. All data were obtained using pClamp 11 acquisition software (Molecular Devices) and analyzed using Easy Electrophysiology software (Easy Electrophysiology Ltd, UK). Only cells with >1 GΩ seal before whole-cell configuration and with series resistance (Rs) < 25 MΩ were recorded. Any recordings with Rs variation greater than 20% between before and after the experiment were excluded from the analysis. We confirmed that there were no significant differences across experimental groups in the following intrinsic property metrics: resting membrane potential, input resistance, and membrane capacitance (Supplementary Fig. 15c).

### Flow cytometry

Brain-infiltrating leukocytes (BILs) were isolated by density-gradient centrifugation using Percoll (GE Healthcare Life Sciences, 17089101), as previously described[52]. After isolation, single-cell suspensions of BILs (0.5 – 1 × $10^6$ cells) were stained with Zombie Aqua™ dye (BioLegend, 423102) to discriminate between live and dead cells. Fc receptor blocking was performed using TruStain FcX PLUS anti-mouse CD16/32 Antibody (BioLegend, 156604) prior to staining with fluorophore-conjugated antibodies at the concentrations recommended by the manufacturers. Fluorescence minus one (FMO) controls were used to determine accurate gating. The antibody panels are listed in Supplementary Data 2a–h. Data were acquired using an Attune NxT flow cytometer (Thermo Fisher Scientific) or Cytek Aurora Spectral Flow Cytometry (Cytek Biosciences) and analyzed using FlowJo software (Tree Star, version 10.8.1).

### Immunosuppression assay via co-culture of TAMs and CFSE-labeled T-cells

CD3+ T-cells were isolated from the spleen of an 8–10-week-old C57BL6/J female healthy donor mouse. The spleen was dissociated into a single-cell suspension, and red blood cells were lysed using ACK lysis buffer (Lonza, 10-548E). CD3+ T-cells were isolated using the Mojosort Mouse CD3 T Cell Isolation Kit (BioLegend, 480031) and labeled with CFSE using the CFSE Cell Division Tracker Kit (BioLegend, 423801); a small fraction of unlabeled T-cells was retained for negative controls. Specifically, 1 μL of CFSE working solution was added to the T-cell suspension ($2 × 10^7$ cells in 2 mL PBS) and incubated at 37 °C for 10 min in the dark. Cells were then washed three times with cRPMI to remove excess dye and resuspended in cRPMI with hIL-2 (30 IU/mL, Pepro-Tech, 200-02). The T-cell suspension was combined with Dynabeads™ Mouse T-Activator CD3/CD28 (Gibco, 11453D) at a 1:1 ratio and plated in a U-bottom 96-well tissue culture plate at $1 × 10^5$ cells per well.

Brain-infiltrating leukocytes (BILs) were isolated from SB28 tumor-bearing mouse brains as described above. Following density-gradient centrifugation and cell counting, CD11b+ myeloid cells were enriched using CD11b MicroBeads, human and mouse (Miltenyi Biotec, 130-049-601).

CFSE-labeled T-cells and CD11b+ BILs were co-cultured at a 1:0.8 ratio (T-cell:CD11b+ cells). The condition of CFSE-labeled T-cells with activation beads (without CD11b+ cells) served as a positive control (PC), while CFSE-labeled T-cells without activation beads served as a negative control (NC). Additional quality control conditions included CFSE-unlabeled T-cells with and without activation beads. After 72 h of co-culture, activation beads were removed using a magnetic stand, and cells were collected from flow cytometric analysis.

### BMDM reprogramming experiment

To obtain bone marrow cells, 6–10-week-old C57BL/6J female mice were euthanized in accordance with established IACUC-approved protocols, and tibias and femurs were harvested in a sterile manner. The cells were flushed with RPMI and dispersed into the culture medium, passed through a 70 μm filter cell strainer (Falcon, 352350), processed with ACK lysis buffer (Lonza, 10-548E), and then passed through a 40 μm-filter cell strainer (Falcon, 352340). The collected cells were resuspended in cRPMI medium supplemented with 20 ng/mL GM-CSF (PeproTech, 315-03) and seeded at a density of approximately $2 × 10^6$ cells per well in 12-well non-TC-treated plates (USA Scientific, CC7672-7512). On day 2, the culture medium was replenished with fresh medium supplemented with 20 ng/mL GM-CSF. On day 5, the culture medium was replaced again with one of the following conditions: (1) cRPMI supplemented with GM-CSF (baseline); (2) cRPMI supplemented with GM-CSF plus IFNγ at 100 ng/mL (PeproTech, 315-05); (3) 1) cRPMI supplemented with GM-CSF plus IL-4 at 20 ng/mL (PeproTech. 214-14)[79]; (4) SB28-TSP1-WT cells; or (5) KO cells ($3 × 10^3$ cells per well) on permeable well inserts with a PET membrane pore size of 0.4 μm (Corning, 353180) (Supplementary Fig. 19a). Co-cultures were maintained for an additional 36 h, after which non-adherent cells were rinsed away by washing with PBS twice. The remaining adherent cells, regarded as bone marrow-derived macrophages (BMDMs), were lysed directly on the culture plates with RLT buffer provided in the Qiagen RNeasy mini kit (Qiagen, 74106).

### Reverse transcription-quantitative PCR (RT-qPCR)

RNA extraction was performed as mentioned above, and cDNA was synthesized using qScript Ultra SuperMix (Quantabio, 95217-100) according to the manufacturer's protocol. RT-qPCR was carried out in 10 μL reactions consisting of 5 μL of 2×PerfeCTa qPCR Supermix ROX II (Quantabio, 95119-012), 0.5 μL of TaqMan probe, 2.5 μL of RNase-free water, and 2 μL of cDNA template, on a MicroAmp™ Optical 384-well reaction plate (Applied Biosystems, 4309849). TaqMan FAM-MGB

probes for the following genes were used: *Nos2*, Mm00440502_m1; *Tnf*, Mm00443258_m1; *Cd80*, Mm0071660_m1; *Cd86*, Mm00444543_m1; *Arg1*, Mm00475988_m1; *Chil4*, Mm00840870_m1; *Retnla*, Mm00445109_m1; *Cd163*, Mm00474091_m1; *Gapdh* Mm99999915 (all from Applied Biosystems). All samples were run in triplicates using the QuantStudio 5 (Applied Biosystems). The quantification cycle ($C_q$) values were exported through QuantStudio Design and Analysis v2 app, and only the data labeled as 'Amp' (= Target amplified) were used for analyses. Relative expression values were calculated using the $2^{-\Delta\Delta Cq}$ method by normalizing the values to *Gapdh* and the corresponding reference sample. The median of technical replicates was used as the representative value for each condition.

### Mouse survival studies

For survival studies, morbidity criteria were consistent with the predetermined IACUC-approved biological endpoint, which included severe physiological symptoms (e.g., hunching, respiratory distress, and general malaise), severe neurological impairments (e.g., circling, ataxia, tremors, seizure, convulsion, head tilt, paralysis, weakness, hunch back, bulge in skull, and balance problems), and a body weight loss of 15% or more from the initial weight. Kaplan–Meier survival analysis, using log-rank testing, was employed to assess statistical significance.

### Mouse drug treatment

For all drug studies, orthotopic tumor cell inoculation was performed as described above. In experiments presented in Fig. 5a, c and Supplementary Figs. 22, 24–26, drug treatment was initiated on the day after tumor inoculation without performing any randomization. Instead, animals were assigned to experimental groups based on body weight, ensuring similar weight ranges across groups. Mice received either perampanel (0.75 mg/kg; Adooq Biosciences, A12498; formulated in 10% DMSO, 60% PEG300, 30% water) or PBS via daily oral gavage. Treatment was continued until reaching predetermined humane endpoints or predetermined tissue harvesting, unless otherwise specified. In experiments presented in Fig. 5b and Supplementary Figs. 23, 27, the following adjustments were made: a control solution (10% DMSO, 60% PEG300, 30% water, 100 μL per administration) replaced PBS and was administered via oral gavage.

### Mouse immunotherapy experiment

Intracranial inoculation of mEGFR-expressing SB28 (TSP1-WT) cells (approximately $1 \times 10^4$ cells in 1 μL sterile HBSS) was performed as described above. On day 7, based on BLI-estimated tumor size, mice were randomized into PER and control groups, with treatment starting on the same day. Anti-PD-1 antibody (BioXCell, #BE0146) was administered intraperitoneally (i.p.) at days 7, 11, 14, 18, 21, and 25, for a total of six doses (200 μg/injection)[70].

On day 8, EGFRvIII-specific CAR T-cells were isolated from the spleen of an EGFRvIII-CAR transgenic mouse as previously described[52–54]. Constitutive expression of EGFRvIII-specific CAR on T-cells was confirmed in the donor mouse in advance by flow cytometry-based testing of peripheral blood samples using PE-conjugated EGFRvIII (ACROBiosystems, EGI-HP2E3). After euthanasia with CO2 inhalation, the spleen was harvested and dissociated into a single cell suspension; red blood cells were lysed using ACK lysis buffer (Lonza, 10-548E). CD3+ T-cells were isolated using the Mojosort Mouse CD3 T Cell Isolation Kit (BioLegend, 480031) following the manufacturer's protocol. The sorted CD3+ T-cells were then co-cultured with Dynabeads™ Mouse T-Activator CD3/CD28 (Gibco, 11453D) at a 1:1 ratio in cRPMI supplemented with IL-2 (30 IU/mL). Activation beads were removed 48 h later using a magnetic stand, and the cells were resuspended in fresh culture media supplemented with IL-2 (30 IU/mL) and IL-15 (50 ng/mL) and maintained for an additional 2 days until infusion.

On day 11, systemic lymphodepletion was achieved through intraperitoneal administration of cyclophosphamide (150 mg/kg) and fludarabine (50 mg/kg). On day 12, CAR T-cells ($1 \times 10^6$ cells per mouse) were resuspended in 100 μL of sterile HBSS and infused retroorbitally into all mice in the cohort. The CAR T-cell infusion product was characterized by flow cytometry on the same day (Supplementary Fig. 27b).

### Data visualization

Graphical illustrations were generated using the BioRender web tool. Data visualization was performed using R (v4.1.2) along with the ggplot2 package (v3.3.6). All figures were combined, readjusted, and finalized in Affinity Designer (Serif, v1.10.5).

### Statistics and reproducibility

Mouse survival analysis was performed using the log-rank test from the R package survival, and Kaplan–Meier survival curves were visualized using the R package *survminer*. A significance level of $p < 0.05$ was considered statistically significant. For two-group comparisons, depending on the normality of data distributions, either the Wilcoxon rank-sum test or Welch's unpaired *t*-test was used, specified in the figure legends. For multiple testing, $p$ values were adjusted using the Benjamini–Hochberg or Bonferroni procedures, as specified in the figure legends, and presented as adjusted $p$ values. Statistical analysis and data visualization were performed using R version 4.1.2. No statistical method was used to predetermine sample size. All experiments were conducted in at least technical duplicates. No data were excluded from the analyses. The experiments were not randomized unless otherwise specified. The investigators were not blinded to allocation during experiments and outcome assessment.

### Reporting summary

Further information on research design is available in the Nature Portfolio Reporting Summary linked to this article.

## Data availability

The 10x Visium mouse spatial transcriptomics data and mouse in vivo tumor model RNA-seq data newly generated in this study are available through the NCBI Gene Expression Omnibus (GEO) website under accession numbers GSE289934 and GSE289935, respectively. The publicly available data used in this study are available as follows: human glioblastoma single-cell RNA-seq: in the GEO database under accession code GSE223065[14]; mouse glioma SB28 RNA-seq: in the GEO database under accession code GSE127075[38]; mouse glioma GL261 RNA-seq: in the GEO database under accession code GSE94239; normal mouse brain bulk RNA-seq: in the EMBL-EBI ArrayExpress database under accession code E-MTAB-6081[72]; 10x Visium mouse brain spatial transcriptomics data of GL261: in the GEO database under accession code GSE245263[40]. Human glioblastoma 10x Visium spatial transcriptomics data are available using the R package SPATAData [https://github.com/theMILOlab/SPATAData][34,35], as described earlier. All other study data are included in the manuscript and/or supporting information. Source data are provided with this paper.

## Code availability

The code used for data analysis is available on Github without any restrictions: https://github.com/t-nejo/Glioma-neuron-immune-crosstalk[80].

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

## Acknowledgements

We thank the following individuals and organizations: the study participants and their families; Anny Shai and Yunita Lim of the UCSF Brain Tumor Center SPORE Biorepository and Pathology Core for their services (NIH/NCI 5P50CA097257-18); Mylinh Bernardi and Horng-Ru Lin of the Gladstone Genomics Core for their assistance with Visium FFPE Spatial Gene Expression assay and sequencing under the support from the James B. Pendleton Charitable Trust for funding Next-Seq 500 sequencer used in this study; Braise Ndjamen of the Gladstone Histology and Light Microscopy Core for 10X Visium Spatial Transcriptomics data acquisition-related services; Anna Celli of the Laboratory for Cell Analysis Core Facility, UCSF Helen Diller Family Comprehensive Cancer Center for her technical assistance related to tissue image acquisition; Lenka Maliskova, Armita Norouzi, and Walter Eckalbar of UCSF Genomics CoLab for the RNA-sequencing library preparation; Sequencing was performed at the UCSF Center for Advanced Technology (CAT) for sequencing, supported by UCSF PBBR, RRP IMIA, and NIH 1S10OD028511-01 grants; the DNA Technologies and Expression Analysis Core at UC Davis Genome Center for bulk RNA-Seq services; members of the S.H.-J. laboratory and H.Okada laboratory for assistance, advice, and helpful discussions. This study was supported by HDFCCC Shared Resource Facilities, Laboratory for Cell Analysis and Preclinical Therapeutics Core, through NIH (P30CA082103), and also supported by HDFCCC NeuroOncology Program to T.N., S.L.H.-J., and H.Okada; JSPS Overseas Research Fellowships (202060725) to T.N.; the UCSF Brain tumor SPORE Career Enhancement Program, NIH grant P50CA097257 to S.K., C.R.C., S.L.H.-J.; Chan-Zuckerberg Biohub Physician-Scientist Fellowship to J.S.Y.; The German Research Foundation DFG GA 3535/1-1 to M.G.; The PhRMA Foundation Post-doctoral Fellowship to A.G.S.D.; NIH grant K08NS126573 and the Sontag Foundation Distinguished Scientist Award to C.R.C.; NIH grants K08NS110919 and R01NS137950, Curci Foundation 7032068 to S.L.H.-J.; NIH grant 1R35 NS105068 and funding from Parker Institute for Cancer Immunotherapy to H.Okada.

## Author contributions

Conceptualization: T.N., S.K., S.L.H.-J., and H.Okada; Formal analysis: T.N. and H.Okada; Funding acquisition: T.N., S.L.H.-J., and H.Okada; Investigation: T.N., S.K., A.Y., S.L., C.J., J.S.Y., T.C., S.S.S.P., L.P., M.G., G.C.M., K.O., D.D., and A.G.S.D.; Methodology: T.N., S.K., A.Y., S.L., M.G., H.Ogino, P.B.W., S.L.H.-J., and H.Okada; Whole-cell patch clamp recording: K.Y.L., D.L.B., and C.R.C.; Calcium imaging: S.K., and S.L.H.-J.; Project administration: T.N., S.K., S.L.H.-J., and H.Okada; Resources: S.K., A.G.S.D., A.C., D.R.R., and S.L.H.-J.; Supervision: S.L.H.-J., and H.Okada; Visualization: T.N.; Writing (original draft): T.N., S.L.H.-J., and H.Okada; Writing (review and editing): T.N., S.K., A.Y., S.L., K.Y.L., D.L.B., J.S.Y., T.C., M.G., P.B.W., C.R.C., D.R.R., S.L.H.-J., and H.Okada.

## Competing interests

The authors declare no competing interests.
