## [Transparent Peer Review file · Nature Communications]

Glioma-neuronal circuit remodeling induces regional immunosuppression

Corresponding Author: Prof Hideho Okada

Version 0:

Reviewer comments:

Reviewer #2

(Remarks to the Author)

I don't really see any changes that have been made that relate to my concerns particularly regarding inadequate animal numbers in survival studies so as to establish that the proposed immuno-redirection alters survival. My second concern related to novelty of the findings. The authors did argue their vantage point that while glutamatergic neuronal hyperexcitability had been shown to promote glioma growth that a redirection of the immune response via neuronal activity had not been demonstrated. I do agree that this is a novel twist yet there are no mechanism whereby this may happen. Smart minds can disagree and I am happy to agree to disagree on the novelty and importance of the findings.

(Remarks on code availability)

Reviewer #3

(Remarks to the Author)

The authors have appropriately addressed the comments made by reviewers and further strengthened the conclusions of their original submission with new, robust experiments. The point by point response and the new experiments were thorough and do not require any further revisions. The results are noteworthy, and have added to our understanding of immunosuppression in glioma. The revised manuscript can be accepted for publication.

(Remarks on code availability)

Reviewer #4

(Remarks to the Author)

In the revised manuscript, the authors have made substantial improvements, adding the following experiments:

- Including re-analysis of published ST data from human patient samples and GL261 mouse models.
- Co-culture assays examining the impact of TAMs on T-cell proliferation.
- Testing the influence of tumor-secreted TSP1 on macrophage polarization.
- Further characterization of CD8+ T-cell populations and their phenotypes.
- Evaluation of neuronal activity by analyzing calcium events in co-culture systems.
- Measurement of spontaneous excitatory postsynaptic currents in ex vivo mouse brain slices.

These additions make the conclusions more reliable. The authors have largely addressed my previous concerns. I recommend the manuscript for acceptance and publication, though I have a few minor questions for the authors to address,

which do not impact my overall recommendation for acceptance.

1. Fig. 1 indicates that the proportion of TSP1+ cells is quite small (~4.9%). Therefore, do the findings in the KO mouse model adequately represent the role of the TSP1 gene in human GBM?
2. In Fig. 1b, the EMT pathway is enriched in myeloid cells. Why? Could this be due to the contamination of mesenchymal cells in the myeloid cell subpopulations during scRNA-seq experiment or analysis?
3. In Fig. 2c, the x-axis represents neuro-synaptic gene signature scores, and the same samples are used in all three scatter plots. Why the x-axis scales differ across the plots despite using the same dataset?
4. In Fig. S5b, the expression of THBS1 does not show a statistically significant difference between HFC and LFC. Therefore, the higher expression of Thbs1 in SB28 tumors alone is insufficient to classify SB28 tumors as HFC-like. More conclusive evidence is needed to justify the validity of the SB28 mouse model. Additionally, a stronger rationale is necessary to define SB28-TSP1-KO tumors as LFC-like. For instance, while TSP1-WT tumors in the mouse model are enriched for Neuronal System and Synaptic Transmission Glutamatergic pathways compared to TSP1-KO tumors, is there similar enrichment observed in HFC and LFC regions in human data?
5. In lines 366–374, it is stated that "CD11b+ cells from TSP1-KO tumors showed significantly less suppressive capacities on T-cell proliferation compared with the WT counterparts." However, there is no data directly proving that the reduction in proliferating T cells is caused by CD11b+ cells. Additionally, could the authors investigate whether similar findings are observed in human scRNA-seq data, specifically comparing the differences in proliferating T cells between HFC and LFC samples?

(Remarks on code availability)

Reviewer #5

(Remarks to the Author)

The authors have submitted an interesting manuscript and have thoroughly addressed all relevant comments. However, I have questions regarding their syngeneic model system:

1. Regarding the SB28 tumor model: Have the authors validated this model (e.g., through sequencing or methylation analysis) to confirm that it accurately represents IDH-WT glioblastoma?
2. Can the authors provide evidence that neuronal activity or neurons contribute to protumorigenic effects in their model system? Further, do they have data supporting synaptic communication between neurons and tumor cells?
3. Are there alternative model systems that could be used in this context? If not, the authors should discuss the limitations of their current model in the discussion section.

(Remarks on code availability)

Version 1:

Reviewer comments:

Reviewer #4

(Remarks to the Author)

The author's point-to-point response and the revised manuscript have thoroughly addressed all my concerns. The data is of high quality and enhances our understanding of the interactions between the nervous system and the glioma immune microenvironment. It is an excellent and highly valuable study. I recommend this manuscript for publication.

(Remarks on code availability)

Dear Editor and Reviewers:

We sincerely appreciate the reviewers' thoughtful and encouraging feedback on our revised manuscript, "Glioma-neuronal circuit remodeling induces regional immunosuppression." We have carefully addressed all comments and provided point-by-point responses below, with the reviewers' comments in *italics*, followed by our responses. Accordingly, we have made the corresponding revisions to the manuscript, which are highlighted in red in the main text and presented in this letter with underlines.

To avoid confusions, we define each version of the manuscript as follows:

Version 1 (v1): Initially submitted to *Nature Cancer* in August 2023 and reviewed by Reviewers 1–4.

Version 2 (v2): Resubmitted to *Nature Cancer* in October 2024, transferred to *Nature Communications* in November 2024, and reviewed by Reviewers 2–5.

Version 3 (v3): The latest version submitted alongside this response letter to *Nature Communications*.

Reviewer #2 (Remarks):

I don't really see any changes that have been made that relate to my concerns particularly regarding inadequate animal numbers in survival studies so as to establish that the proposed immuno-redirection alters survival. My second concern related to novelty of the findings. The authors did argue their vantage point that while glutamatergic neuronal hyperexcitability had been shown to promote glioma growth that a redirection of the immune response via neuronal activity had not been demonstrated. I do agree that this is a novel twist yet there are no mechanism whereby this may happen. Smart minds can disagree and I am happy to agree to disagree on the novelty and importance of the findings.

Response: We sincerely thank the reviewer once again for providing critical and valuable feedback. We recognize that differing perspectives, particularly on novel findings, are inevitable and can enrich scientific discourse. While we respect the reviewer's viewpoint, we remain confident in the originality and significance of our study, as also acknowledged by the other four reviewers.

Regarding the first point on "inadequate animal numbers in survival studies," we would like to clarify that we reorganized the study to focus on mechanistic investigations, removing most of the mouse survival data in the previously revised manuscript (**v2**) to better align with the primary objectives of our research. We provided this clarification at the beginning of our previous point-by-point response letter. The only two survival studies that remain in the current manuscript (**v2**, **v3**) are: (1) the assessment of PER's effect (**Fig. 4h**; $n = 10$ per group) and (2) the evaluation of CAR-T and anti-PD1-Ab combination therapy, with and without PER (**Extended Data Fig. S27c**; $n = 9$ per group). Compared to similar mouse survival experiments in published studies (*Krishna S, 2023 Nature; Venkatesh HS, 2019 Nature*)^{1,2}, we believe our sample sizes are appropriate and do not constitute a major limitation. We hope the reviewer understands this rationale and, therefore, that the concern may not be fully applicable to the current manuscript (**v2**, **v3**).

On the second point, we have revised the Discussion section in the latest version of the manuscript (v3) to further highlight the novelty and significance of our findings, ensuring the readers can readily appreciate these aspects.

(Page 30, Line 1) Advancements in cancer neuroscience research have revealed unique features and treatment resistance mechanisms specific to brain tumors¹⁻⁹. Furthermore, contributions of the immune axis to neurofibromatosis-1 (NF1) low-grade glioma growth have also been investigated¹⁰. More recently, Drexler et al. epigenetically defined the neural signature of glioblastoma and demonstrated its anticorrelation with the immune component¹¹. Building on this knowledge, our interdisciplinary investigation integrates cancer neuroscience, cancer immunology, and neuroimmunology to examine how glioblastoma remodels neuronal circuits to evade immune surveillance.

In this study, we uncover a previously unrecognized process in which glioma-neuronal interactions contribute to regional immunosuppression within the cortex remodeled by glioblastoma infiltration. Our findings fill the gap between the well-established concepts of cancer neuroscience^{1-9,12-15} and glioma-associated immunosuppression¹⁶⁻¹⁸. Expanding on the foundational knowledge established by Krishna et al.² regarding TSP1's role in glioma-neuronal circuit interactions, we mechanistically demonstrate that TSP1-expressing glioblastoma regions with HFC exhibit significant suppression of key immune response pathways, accompanied by an increase in immunosuppressive TAMs (Fig. 4i). Spatial transcriptomic analysis in human cases and preclinical glioblastoma models reveals an inverse relationship between neuronal activity-related and inflammatory response signatures. Further, our SB28-TSP1-WT/KO model determines the role of TSP1 in modulating both the cellular composition and immune functionality of the tumor microenvironment. These findings highlight the pivotal role of glioma-neuronal circuit remodeling as a critical driver of immune evasion in hyperexcitable glioblastomas, corroborating recent findings from other studies¹⁵.

Beyond these mechanistic insights, our study advances the experimental models for investigating glioma-neuronal-immune interactions. By leveraging the syngeneic SB28 mouse glioma model and integrating *in vitro*, *in vivo*, and *ex vivo* systems, we provide a framework to dissect these interactions under controlled conditions. Moreover, our findings have direct therapeutic implications, as we show that inhibiting glutamatergic signaling can reverse immunosuppression, potentially enhancing the efficacy of immunotherapies. While previous studies have explored the role of perampanel (PER) and other glutamate receptor antagonists in glioma-neuron interactions¹⁹, our study is among the first to link these treatments with immune modulation^{20,21}. Although our preclinical results did not reach statistical significance in combination therapy settings, they lay the groundwork for future investigations into potential synergy between anti-epileptic drugs and immunotherapy. Given the parallels between our findings and clinical observations^{22,23}, targeting glioma-neuronal crosstalk may represent a promising avenue for therapeutic intervention.

Reviewer #3 (Remarks):

The authors have appropriately addressed the comments made by reviewers and further strengthened the conclusions of their original submission with new, robust experiments. The point by point response and the new experiments were thorough and do not require any further revisions. The results are noteworthy, and have added to our understanding of immunosuppression in glioma. The revised manuscript can be accepted for publication.

Response: We appreciate the reviewer for the careful review of our point-by-point response and the revised manuscript, and encouraging comments.

Reviewer #4 (Remarks):

In the revised manuscript, the authors have made substantial improvements, adding the following experiments:

- *Including re-analysis of published ST data from human patient samples and GL261 mouse models.*
- *Co-culture assays examining the impact of TAMs on T-cell proliferation.*
- *Testing the influence of tumor-secreted TSP1 on macrophage polarization.*
- *Further characterization of CD8+ T-cell populations and their phenotypes.*
- *Evaluation of neuronal activity by analyzing calcium events in co-culture systems.*
- *Measurement of spontaneous excitatory postsynaptic currents in ex vivo mouse brain slices.*

These additions make the conclusions more reliable. The authors have largely addressed my previous concerns. I recommend the manuscript for acceptance and publication, though I have a few minor questions for the authors to address, which do not impact my overall recommendation for acceptance.

Response: Thank you for evaluating our extensive revision work fairly and thoroughly.

Reviewer 4 Comment 1: *Fig. 1 indicates that the proportion of TSP1+ cells is quite small (~4.9%). Therefore, do the findings in the KO mouse model adequately represent the role of the TSP1 gene in human GBM?*

Response: We believe this question pertains to **Fig. 1i**, which was added in response to a suggestion from Reviewer 4 during the previous review cycle (**Reviewer 4 Comment 5**). These data, now presented in **Fig. 1i** and **Extended Data Fig. S5**, evaluate TSP1 expression specifically in the myeloid cell population.

We explored the difference in TSP1 expression between HFC- and LFC-TAMs within the myeloid cell population and found that Mo-TAMs, which are more enriched in HFC-TAMs, may contribute to creating the immunosuppressive tumor microenvironment by secreting TSP1, albeit to a lesser extent than tumor cells (**Fig. 1i; presented below**).

Importantly, we would like to emphasize that tumor cells are the primary source of TSP1 in the HFC region. Our previous study (**Krishna, Hervey-Jumper, et al., 2023 Nature**) showed that tumor cells within the HFC region expressed significantly higher levels of TSP1 than those in the LFC region. Moreover, the study demonstrated that tumor-secreted TSP1 induces key features of the HFC region, including greater integration into neuronal organoids, more frequent synapse formation, tumor hyperexcitability, enhanced proliferation, and increased aggressiveness *in vivo*. These findings provided the mechanistic rationale for developing the TSP1 knockout mouse model used in our present study.

Reviewer 4 Comment 2: *In Fig. 1b, the EMT pathway is enriched in myeloid cells. Why? Could this be due to the contamination of mesenchymal cells in the myeloid cell subpopulations during scRNA-seq experiment or analysis?*

Response: Thank you for bringing this point to our attention. During the previous review cycle, in response to the question from Reviewer 4 (**Reviewer 4 Comment 1**), we conducted a copy-number variation (CNV) imputation analysis with our human single-cell RNA-seq data. The analysis confirmed that CNVs, such as gain of chromosome 5 or 7 and loss of chromosome 10, were exclusively observed in tumor cell population (**a–d**). These findings indicate that contamination of mesenchymal cells in the myeloid cell subpopulations is unlikely.

Additionally, we carefully examined the “epithelial-mesenchymal-transition (EMT) pathway” gene set in the Hallmark collection. In this gene set, we found that many macrophage-relevant genes were included in the leading-edge genes upregulated in HFC-TAMs (e.g., *VIM*, *AREG*, *LGALS1*, *THBS1*, *CD44*, *TGFBI*, *SPPI1*, *ITGA5*, *ITGAV*, and *MMP14*) (**e**). We presume that the upregulation of these genes likely contributed to the observed enrichment of the EMT pathway.

Figure related to Reviewer 4 Comment 2 |

a–d, CNV analysis to validate cell annotations of tumor and non-tumor cells. Heatmap showing chromosomal-level copy number variations (CNVs) identified in tumor cells within the human glioblastoma sc-RNA-seq dataset. According to the original cell annotations reported by Krishna et al., all non-tumor cells were used as references. In the heatmap, red and blue indicate inferred chromosomal gain and loss, respectively (**a**). Feature plots showing the distribution of each patient's cells (**b**), the distributions of gain of chromosome 7 (**c**) and loss of chromosome 10 (**d**).

e, GSEA enrichment plot of the Hallmark Epithelial-Mesenchymal Transition gene set, comparing myeloid cell populations derived from HFC and LFC samples. Among the 47 identified leading-edge genes, representative "myeloid cell-associated" genes are highlighted.

Reviewer 4 Comment 3: *In Fig. 2c, the x-axis represents neuro-synaptic gene signature scores, and the same samples are used in all three scatter plots. Why the x-axis scales differ across the plots despite using the same dataset?*

Response: Thank you for raising this point. We believe the question pertains to the y-axis, which represents neuro-synaptic gene signature scores, rather than the x-axis. We confirm that the values for the neuro-synaptic gene signature scores are identical across all three scatter plots. However, the y-axis scale in the middle panel was adjusted due to the distribution of the regression line, which caused the scales to appear inconsistent. In the latest revised manuscript (v3), we have corrected this in **Fig. 2c**, as well as in **Extended Data Fig. S6** to ensure consistent y-axis scaling across all panels, enhancing clarity and visual uniformity.

Reviewer 4 Comment 4: *In Fig. S5b, the expression of THBS1 does not show a statistically significant difference between HFC and LFC. Therefore, the higher expression of Thbs1 in SB28 tumors alone is insufficient to classify SB28 tumors as HFC-like. More conclusive evidence is needed to justify the validity of the SB28 mouse model. Additionally, a stronger rationale is necessary to define SB28-TSP1-KO tumors as LFC-like. For instance, while TSP1-WT tumors in the mouse model are enriched for Neuronal System and Synaptic Transmission Glutamatergic pathways compared to TSP1-KO tumors, is there similar enrichment observed in HFC and LFC regions in human data?*

Response: Regarding the first point, the data presented in **Fig. S5** and **Fig. 1i** pertain to the myeloid cell populations, not tumor cell populations. As we noted earlier in **our response to Reviewer 4 Comment 1**, while there is no statistically significant difference in TSP1 expression between HFC- and LFC-derived myeloid cells (or TAMs), this observation does not contradict previous findings, since tumor cells are identified as the primary source of TSP1 expression in HFC intratumoral regions, as reported by **Krishna, Hervey-Jumper, et al. (2023 Nature)**.

Regarding the second concern about the validity of SB28-TSP1-KO tumors as a model for LFC-like tumors, we provided additional supporting evidence in the manuscript (**v2, v3**), including immunofluorescence synapse imaging, live-cell calcium imaging, and whole-cell patch clamp recordings in acute brain slices. These data consistently demonstrate that SB28-TSP1-KO tumors make fewer synaptic contacts in neuron-glioma co-culture and exhibit less frequent excitatory activity compared to their SB28-TSP1-WT counterparts, thereby supporting the transcriptomic and electrophysiological similarities of SB28-TSP1-KO tumors to human glioblastoma LFC regions.

Regarding the final point, we conducted a transcriptomic analysis comparing HFC- and LFC-derived neurons in human single-cell RNA-seq data, using the same approach as in **Fig. 1a-c**. The table below summarizes the top pathways (adjusted $P < 0.2$) from the KEGG, Reactome, GO:BP, and Hallmark gene set collections. Positive (red) and negative (blue) NES values indicate upregulation and downregulation in HFC relative to LFC, respectively. The analysis suggests upregulation of neuro-synaptic activity in HFC compared to LFC, which aligns with our observations from bulk RNA-seq of *in vivo* tumors comparing SB28-TSP1-WT and KO. Although the limited number of neurons in the dataset (e.g., only 60 and 2 neurons in patient 2) prevents us from drawing definitive conclusions, HFC and LFC intratumoral regions, by definition, differ in neuro-synaptic connectivity, as evidenced by the increased frequency and amplitude of excitatory synaptic signals in HFC regions. We anticipate that these differences may also be reflected at the transcriptomic level, for example, in *Neuronal System* and *Synaptic Transmission Glutamatergic* pathways, though this remains to be validated.

Table: Summary of GSEA comparing HFC and LFC-derived neurons.

no	pathway	padj	NES	leadingEdge
1	REACTOME_NA_CL_DEPENDENT_NEUROTRANSMITTER_TRANSPORTERS	0.0006	1.744	SLC6A6, SLC6A1, SLC6A15
2	GOBP_REGULATION_OF_POSTSYNAPTIC_NEUROTRANSMITTER_RECEPTOR_ACTIVITY	0.033	1.602	NPTX2, AKAP9, NPTXR, DLGAP4
3	GOBP_NEUROTRANSMITTER_UPTAKE	0.155	1.568	GLUL, SLC38A1, ITGB1, GDNF, KCNJ10, NOS1, SNCA, GPM6B
4	GOBP_NEURON_PROJECTION_EXTENSION_INVOLVED_IN_NEURON_PROJECTION_GUIDANCE	0.079	1.565	VEGFA, NRP2, SEMA3C, SEMA6A, SEMA6D, SEMA4D, SEMA3B, SEMA7A, RYK
5	GOBP_SYNAPTIC_VESICLE_ENDOSOMAL_PROCESSING	0.125	1.495	EEA1, BTBD8, VAMP4, AP3D1, ITSN1, GRIPAP1, AP3B1
6	GOBP_COMMISSURAL_NEURON_AXON_GUIDANCE	0.124	1.483	VEGFA, NFIB, GDNF, NCAM1, FZD3, RYK
7	GOBP_NEGATIVE_REGULATION_OF_NEURON_PROJECTION_DEVELOPMENT	0.107	1.448	LGALS1, VIM, DNM3, SPOCK1, MT3, INPP5F, NLGN1, RUFY3, DIP2B, LRIG2, SEMA3C, EFN2, SEMA6A, SEMA6D, SEMA4D, SEMA3B, CBFA2T2, LINGO1, NGFR, TRAK2, CERS2, LRP1, OMG, ARHGDI2, FLNA, SPP1, BAG5, SEMA7A, RYK, MYLIP, GSK3A, KREMEN1, PAQR3
8	GOBP_NEURON_MIGRATION	0.154	1.399	VEGFA, NRP2, DNER, SPOCK1, MAP1B, NRCAM, TUBB2B, NAV1, FBXO45, ADGRG1, LRIG2, SH3RF1, NTRK2, ZMIZ1, SEMA6A, USP9X, PCM1, CDKL5, DAB2IP, FGFR1, ADGRL3, NIPBL, CRKL, PAFAH1B1, FLNA, ACAP3, SHTN1, PLAA, FZD3, MARK2, PHACTR1, RAPGEF2, CTNNA2
9	GOBP_NEURON_PROJECTION_GUIDANCE	0.055	1.385	VEGFA, NRP2, NRCAM, SCN1B, ROBO1, TUBB2B, APP, ENAH, PIK3CB, NFIB, PTPN11, GAP43, SEMA3C, EFN2, SEMA6A, VLDLR, SEMA6D, SEMA4D, GDNF, NCAM1, SEMA3B, NFASC, PLXND1, MYOT, PRKCA, NRXN3, RPS6KA5, MAPK1, NGFR, LRP1, TRIO, KIF5A, FRS2, PTPRA, PTPRM, KIF5B, SPTBN1, KLF7, MYCBP2, SHANK3, SPTAN1, SH3KBP1, PLCG1, PLXNB1, RAP1GAP, FZD3, SMAD4, PRKQC, LAMB2, KIF5C, SEMA7A, PIK3R1, RYK, LAMA2, ZSWIM6
10	GOBP_REGULATION_OF_NEURON_PROJECTION_DEVELOPMENT	0.052	1.359	LGALS1, VIM, VEGFA, DNM3, SPOCK1, MAP1B, MT3, NEDD4L, MACF1, INPP5F, SCN1B, NLGN1, ROBO1, PREX1, RUFY3, CFLAR, DIP2B, GOLGA4, LRIG2, ADAM10, SEMA3C, FUT9, NTRK2, EFN2, SS18L2, SEMA6A, VLDLR, SEMA6D, SEMA4D, SRCIN1, CCDC88A, TIAM1, COBL, HSPA5, SEMA3B, CBFA2T2, CDKL5, FNI, NFE2L2, FBXO38, PLXND1, DAB2IP, ARC, LINGO1, MAP2K1, NGFR, RND2, KLF4, PDLIM5, KIF1A, TRAK2, CERS2, LRP1, AKT1, ACTR2, NDRG4, NCS1, OMG, NEDD4, CRKL, FBXO7, PAK3, OPA1, DGKG, TMEM30A, ARHGDI2, FLNA, SHANK3, ACAP3, SHTN1, PRKCI, PTPRD, PLXNB1, SPP1, IST1, KATNB1, KIF13B, MARK2, FBXW8, ITGA6, BAG5, SEMA7A, RAPGEF2, PPFIA2, IQGAP1, CDH2, EZH2, ATF1, CTNNA2, HECW2, RYK, ZDHHC15, MFSD2A, MYLIP, NME1, GSK3A, KREMEN1, CAPRIN2, CNR1, PAQR3, RTN4IP1, SS18L1, S100A9, MDK, SDC2, STYXL1, TNR, THY1, TIAM2, RRM3, RAPGEF1
11	KEGG_NEUROACTIVE_LIGAND_RECEPTOR_INTERACTION	0.101	-1.307	GRID2, THRA, NPY2R, GRIA4, TSPO, C5AR1, C3AR1, ADRA2B, P2RY13, LPAR6, ADORA3, ADRB2, GRIA3, PTGER4, NPFRR1, GABRA2
12	GOBP_NEURON_APOPTOTIC_PROCESS	0.124	-1.343	CCL3, JUN, MCL1, APOE, GRID2, BTG2, NR4A2, HSP90AB1, TYROBP, HRAS, CEBPB, MAG, BOK, CX3CR1, C5AR1, MEF2C, PTPRZ1, AXL, TNF, CASP2, RETREG1, CCL2, CORO1A, ATN1, EGLN3, PINK1, LGMN, MAP3K11, SET, ARRB2, UNC5B, PIN1, RASA1, BAX
13	GOBP_NEURON_DEATH	0.059	-1.363	CCL3, JUN, EGR1, FOS, C1QA, MCL1, APOE, GRID2, UBB, BTG2, NR4A2, HSP90AB1, NR4A3, TYROBP, HRAS, CEBPB, MAG, BOK, CX3CR1, DHCR24, C5AR1, MEF2C, PTPRZ1, AXL, TNF, CASP2, RETREG1, CCL2, CORO1A, ATN1, EGLN3, PINK1, DCC, SLC25A27, LGMN, SLC7A11, MAP3K11, CSF1, SET, ARRB2, TNFRSF1B, REL, UNC5B, PIN1, HTRA2, RASA1, BAX, CTSZ, TLR4, SARM1, BID, ITGB2, RACK1, UBE2M, DAXX, CDC42, TOX3, NSMF, TSC1, SIRT1, AIMP2, KCNB1, RB1, BAD, WNT5A, ALKBH1, TMEM259, BCL2L11
14	GOBP_POSITIVE_REGULATION_OF_NEURON_DEATH	0.176	-1.476	CCL3, JUN, EGR1, FOS, C1QA, MCL1, TYROBP, TNF, CASP2, MAP3K11, PIN1, BAX, TLR4, SARM1, ITGB2, UBE2M, DAXX, CDC42, AIMP2, BAD, WNT5A, BCL2L11, CDC34, REST
15	GOBP_POSITIVE_REGULATION_OF_NEURON_APOPTOTIC_PROCESS	0.186	-1.483	CCL3, JUN, MCL1, TYROBP, TNF, CASP2, MAP3K11, PIN1, BAX, UBE2M, CDC42
16	GOBP_POSITIVE_REGULATION_OF_NEUROINFLAMMATORY_RESPONSE	0.049	-1.716	CCL3, IL1B, TREM2, IL18, CTSC, TNF
17	GOBP_SYNAPSE_PRUNING	0.012	-1.742	C1QB, C1QA, C1QC, TREM2, C3, CX3CR1
18	GOBP_NEUROINFLAMMATORY_RESPONSE	0.074	-1.748	CCL3, IL1B, TREM2, PTPRC, IL18, CTSC, TNF, TNFRSF1B, PTGS2

Reviewer 4 Comment 5: *In lines 366–374, it is stated that "CD11b+ cells from TSP1-KO tumors showed significantly less suppressive capacities on T-cell proliferation compared with the WT counterparts." However, there is no data directly proving that the reduction in proliferating T cells is caused by CD11b+ cells. Additionally, could the authors investigate whether similar findings are observed in human scRNA-seq data, specifically comparing the differences in proliferating T cells between HFC and LFC samples?*

Response: We appreciate the reviewer’s feedback. Regarding the first concern, we regret that our presentation of the data may have been unclear. In **Fig. 4d**, the experimental design was structured such that the only variable between the WT/KO samples and the positive control (labeled as “PC”) was the presence or absence of CD11b+ cells isolated from tumor-bearing mouse brains, respectively. As detailed in the **Method** section (cited below), the positive control consisted of CFSE-labeled CD3+ T cells derived from the same healthy donor mouse and

activated using CD3/CD28 activation beads, without the addition of CD11b⁺ cells. In contrast, the test co-culture conditions included CD11b⁺ cells isolated from either TSP1-WT or KO tumors. Given this setup, we believe it is reasonable to conclude that the reduction in T-cell proliferation was directly attributable to the presence of CD11b⁺ cells. To improve clarity, we revised the corresponding text in the manuscript as seen below.

Regarding the second point, in response to the reviewer's suggestion, we reanalyzed the lymphoid cell subset in our human single-cell RNA-seq dataset (86 cells from HFC and 309 cells from LFC regions). Using Seurat's *CellCycleScoring()* function, we calculated cell-cycle scores (S and G2M scores) and estimated the cell-cycle phase (G1, S, or G2M) for each cell. Our analysis revealed a slight trend of higher S and G2M scores in LFC-derived cells compared to HFC-derived cells, which was further supported by a statistically significant difference in the proportion of cells in non-G1 phases between the two regions (Fisher's exact test, $P = 0.026$).

While these preliminary findings appear to align with our functional assay data, we acknowledge several limitations: the observed differences are modest, the number of cells analyzed is relatively small, and the relevance of these cell-cycle scores to our experimental observations has not been validated. As such, we choose not to include these preliminary findings in the revised manuscript but intend to explore this further in future studies.

(Page 53 Line 5 – Method section)

... The T-cell suspension was combined with Dynabeads™ Mouse T-Activator CD3/CD28 (Gibco, 11453D) at a 1:1 ratio and plated in a U-bottom 96-well tissue culture plate at 1×10^5 cells per well.

...
CFSE-labeled T-cells and CD11b⁺ BILs were co-cultured at a 1:0.8 ratio (T-cell:CD11b⁺ cells). The condition of CFSE-labeled T-cells with activation beads (without CD11b⁺ cells) served as a positive control (PC), while CFSE-labeled T-cells without activation beads served as a negative control (NC). Additional quality control conditions included CFSE-unlabeled T-cells with and without activation beads. After 3 days of co-culture, activation beads were removed using a magnetic stand, and cells were collected from flow cytometric analysis.

Figure related to Reviewer 4 Comment 5 |

a, Violin plots showing the distribution of Seurat-calculated S scores and G2M scores in lymphoid cells derived from HFC and LFC regions.

b-c, Bar plots depicting the count (**b**) and fraction (**c**) of cells assigned to G1, S, and G2M phases.

Reviewer #5 (Remarks):

The authors have submitted an interesting manuscript and have thoroughly addressed all relevant comments. However, I have questions regarding their syngeneic model system:

Response: Thank you for your thoughtful review of our manuscript and our responses to the previous reviewers' comments. We sincerely appreciate the time and effort you have dedicated to providing valuable feedback.

Reviewer 5 Comment 1: *Regarding the SB28 tumor model: Have the authors validated this model (e.g., through sequencing or methylation analysis) to confirm that it accurately represents IDH-WT glioblastoma?*

Response: The SB28 tumor model has been extensively characterized in our previous studies (*Genoud V, 2017 OncoImmunology; Simonds EF, 2021 JITC*)²⁴. The cell line was originally developed in our laboratory as a *de novo* glioma in a C57BL/6J mouse by intraventricular transfection of PDGF/shp53/n-Ras using Sleeping-Beauty transposon technology²⁵⁻³³. Please note that we have deposited the SB28 cell line at the Leibniz Institute DSMZ, German Collection of Microorganisms and Cell Cultures GmbH. To date, over 150 laboratories worldwide have obtained the cell line for glioblastoma (GBM). Many groups have used this cell line and described the cell line as one of the most clinically relevant GBM cell lines in mice³⁴. Below, we summarize key characteristics of the SB28 glioma cell line.

- (1) SB28 harbors a mutation load comparable to that of human GBM, with approximately 100 non-synonymous mutations, while the GL261 line, which has been widely used historically, has over 4,000 non-synonymous mutations²² **(a)**.
- (2) The *in vivo* tumor exhibits a highly invasive growth pattern, with tumor cells intermingling with various non-tumor cells at the invading edge **(b)**.
- (3) The orthotopic tumor is heavily infiltrated by myeloid cells, primarily tumor-associated macrophages, and is characterized by a TGF β -high immunosuppressive tumor microenvironment **(b, c)**.
- (4) Consequently, SB28 tumors demonstrate strong resistance to immune checkpoint blockade therapy **(d)**.

These genetic and phenotypic characteristics closely recapitulate those of human IDH-WT GBM, distinguishing SB28 from other syngeneic models, such as GL261²². Therefore, we believe that SB28 represents a particularly suitable GBM model for investigating immunotherapeutic strategies.

[REDACTED]

Figure related to Reviewer 5 Comment 1 |

a, (Adapted from Figure 2 of Genoud V, et al. 2018 *OncoImmunology*) Circos plots depicting the total numbers of non-synonymous somatic mutations (missense or frameshift) in SB28 and GL261 glioma models. The outer circles represent pathways affected by these mutations.

b, Confocal images of immunofluorescence staining of an *in vivo* SB28 tumor. Green: GFP-expressing SB28 cells; Red: TGF β ; Cyan: Iba-1; Blue, DAPI. Scale bars, 100 μ m.

c, (Adapted from Figure 1E of Simonds EF, et al. 2021 *JITC*) Scatter plots showing CyTOF single-cell measurements of CD11b and CD3e in dissociated CD45+ cells from GL261 or SB28 tumors.

d, (Adapted from Figure 5 of Genoud V, et al. 2018 *OncoImmunology*) Kaplan-Meier curves depicting symptom free survival of mice implanted with 1,600 SB28 glioma cells (top) or 50,000 GL261 glioma cells (bottom).

Reviewer 5 Comment 2: *Can the authors provide evidence that neuronal activity or neurons contribute to protumorigenic effects in their model system? Further, do they have data supporting synaptic communication between neurons and tumor cells?*

Response: Regarding the first point—whether neuronal activity or neurons contribute to protumorigenic effects in our model system—we addressed this by co-culturing SB28 cells with neurons and quantifying tumor cell proliferation based on Ki-67 expression in SB28 cells. This experiment followed the same approach used by **Krishna, Hervey-Jumper, et al., 2023 Nature**². Specifically, SB28 cells were co-cultured with mouse cortical neurons for 24 hours on coverslips and then subjected to immunofluorescence staining for Ki-67. The fraction of Ki-67-positive nuclei was quantified in GFP-expressing SB28 cells, revealing that SB28 cell proliferation was significantly increased in co-culture with neurons compared to monoculture conditions (mean, 0.57 vs. 0.37%; Welch’s unpaired *t*-test $P = 1.2 \times 10^{-5}$). These new data have been incorporated as **Extended Data Fig. S9** and the following sentence has been added.

(Page 16, Line 15) Additionally, while SB28 tumor cells do not require neuronal trophic factors for proliferation, they do exhibit significantly increased proliferation when co-cultured with mouse cortical neurons (mCN) compared with SB28 monoculture (mean proliferation index: 0.37 [SB28 alone] vs. 0.57 [SB28 + mCN]; $P = 1.2 \times 10^{-5}$) (Extended Data Fig. S9).

Figure related to Reviewer 5 Comment 2 |

a, Immunofluorescence images of SB28 cells cultured alone (top) or co-cultured with mouse cortical neurons (mCN) (bottom). Green: GFP-expressing SB28 cells; Red: Ki-67; Blue: DAPI. Scale bars, 50 μm .

b, Bar plot quantifying the proliferation index, calculated as the number of Ki67+ nuclei divided by the number of DAPI+ nuclei within GFP-expressing SB28 cells under each condition. Data were obtained from 30 randomly selected fields of view (FOVs) per group (663 $\mu\text{m} \times 663 \mu\text{m}$ each), across three independent culture experiments (10 FOVs per group per experiment). Mean proliferation index: 0.37 (SB28) vs. 0.57 (SB28+mCN) (Welch’s unpaired *t*-test, $P = 1.2 \times 10^{-5}$). Data are presented as mean \pm s.e.m.

For the second point—whether synaptic communication occurs between neurons and tumor cells—this is indeed an intriguing question. In this study, our primary focus has been on how glioma cells influence neuronal activity and the subsequent effects on the immune tumor microenvironment. However, understanding the bidirectional communication between neurons and tumor cells is of great mechanistic interest in the context of our SB28 preclinical model.

To investigate this, we analyzed confocal immunofluorescence images from 24-hour co-cultures of mouse cortical neurons (mCN) and SB28-TSP1-WT cells, in the same series presented in **Fig. 3b**. Specifically, we examined whether co-localized pre- and post-synaptic puncta could be detected, where Synapsin-1 (a presynaptic marker) is expressed in neurons and Homer-1 (a postsynaptic marker) is expressed in tumor cells. While abundant neuron-to-neuron synapses were observed, glioma-to-neuron synapses were hardly detected (**c**). A small number of co-localized puncta, potentially representing glioma-to-neuron synapses, were identified at the edges of GFP-positive areas (**indicated by white arrowheads in the higher magnification images**). However, distinguishing true glioma-to-neuron synaptic structures from overlapping neuron-to-neuron synapses remains technically challenging. Even if these are genuine synaptic structures, their low abundance suggests that direct synaptic communication between glioma and neurons is likely minimal after 24 hours of co-culture.

Nonetheless, as presented in **Fig. 3c, d**, our calcium imaging experiments demonstrate that SB28-TSP1-WT cells modulate the surrounding neuronal circuit, although the effect did not reach statistical significance. The differential impact of SB28-TSP1-WT versus SB28-TSP1-KO supports the role of TSP1 secreted by SB28 cells in neuronal modulation. TSP1 may influence both neuron-to-neuron and glioma-to-neuron synapse formation, raising the possibility that paracrine signaling, rather than direct synaptic communication, plays a predominant role in this context.

At present, apart from the observed increase in tumor cell proliferation, the precise mechanisms by which neuronal systems influence the SB28 glioma model remain unclear. Further comprehensive investigations will be necessary to fully elucidate this aspect. We have added the following statement in the **Discussion** section to acknowledge this as a limitation.

(Page 31, Line 20) Further investigations are necessary to explore these hypotheses and to elucidate the molecular mechanisms underlying the bidirectional communication between glioma cells and neurons, which regulate the recruitment and phenotype of immune cells, including TAMs.

Figure related to Reviewer 5 Comment 2 (continued) |

c, Representative confocal images of mouse neonatal cortical neurons (mCN) co-cultured with SB28-TSP1-WT cells for 24 hours. Images are shown at a $135 \mu\text{m} \times 135 \mu\text{m}$ field-of-view (FOV) (left) and at higher magnification (right), with magnified regions indicated by white rectangles. Green, synapsin-1 (pre-synaptic puncta); red, Homer-1 (post-synaptic puncta); white, MAP2 (neurons); cyan, GFP (SB28 tumor cells); blue, DAPI. In the higher magnification images, GFP-positive areas are outlined in cyan instead of showing direct GFP signals. Co-localized synaptic puncta detected at the edges of GFP-positive areas are indicated by white arrowheads. Scale bar, $20 \mu\text{m}$.

Reviewer 5 Comment 3: *Are there alternative model systems that could be used in this context? If not, the authors should discuss the limitations of their current model in the discussion section.*

Response: Thank you for raising this important point. As discussed earlier (**Reviewer 5 Comment 1**), while the GL261 is also available in our lab, we consider the SB28 and model as the most clinically relevant model available to us. Notably, the use of a syngeneic and poorly immunogenic mouse model enabled us to mechanistically investigate glioma-neuron-immune interactions under controlled conditions. However, we fully acknowledge the limitations of this approach and have addressed them in the **Discussion** section with the following description.

(Page 31, Line 18) Additionally, the exclusive use of SB28 represents an essential limitation of this study, and future investigations employing additional syngeneic models will help strengthen and expand our findings.

References

1. Venkatesh, H. S. *et al.* Electrical and synaptic integration of glioma into neural circuits. *Nature* **573**, 539–545 (2019).
2. Krishna, S. *et al.* Glioblastoma remodelling of human neural circuits decreases survival. *Nature* **617**, 599–607 (2023).
3. Venkatesh, H. S. *et al.* Neuronal Activity Promotes Glioma Growth through Neuroligin-3 Secretion. *Cell* **161**, 803–816 (2015).
4. Venkatesh, H. S. *et al.* Targeting neuronal activity-regulated neuroligin-3 dependency in high-grade glioma. *Nature* **549**, 533–537 (2017).
5. Venkataramani, V. *et al.* Glutamatergic synaptic input to glioma cells drives brain tumour progression. *Nature* **573**, 532–538 (2019).
6. Pan, Y. *et al.* NF1 mutation drives neuronal activity-dependent initiation of optic glioma. *Nature* **594**, 277–282 (2021).
7. Venkataramani, V. *et al.* Glioblastoma hijacks neuronal mechanisms for brain invasion. *Cell* (2022) doi:10.1016/j.cell.2022.06.054.
8. Chen, P. *et al.* Olfactory sensory experience regulates gliomagenesis via neuronal IGF1. *Nature* **606**, 550–556 (2022).
9. Huang-Hobbs, E. *et al.* Remote neuronal activity drives glioma progression through SEMA4F. *Nature* (2023) doi:10.1038/s41586-023-06267-2.
10. Guo, X. *et al.* Midkine activation of CD8⁺ T cells establishes a neuron-immune-cancer axis responsible for low-grade glioma growth. *Nat. Commun.* **11**, 2177 (2020).
11. Drexler, R. *et al.* A prognostic neural epigenetic signature in high-grade glioma. *Nat. Med.* (2024) doi:10.1038/s41591-024-02969-w.
12. Buckingham, S. C. *et al.* Glutamate release by primary brain tumors induces epileptic activity. *Nat. Med.* **17**, 1269–1274 (2011).
13. Campbell, S. L., Buckingham, S. C. & Sontheimer, H. Human glioma cells induce hyperexcitability in cortical networks. *Epilepsia* **53**, 1360–1370 (2012).
14. Campbell, S. L. *et al.* GABAergic disinhibition and impaired KCC2 cotransporter activity underlie tumor-associated epilepsy. *Glia* **63**, 23–36 (2015).
15. John Lin, C.-C. *et al.* Identification of diverse astrocyte populations and their malignant analogs. *Nat. Neurosci.* **20**, 396–405 (2017).

16. Nejo, T., Mende, A. & Okada, H. The current state of immunotherapy for primary and secondary brain tumors: similarities and differences. *Jpn. J. Clin. Oncol.* **50**, 1231–1245 (2020).
17. Chuntova, P. *et al.* Unique challenges for glioblastoma immunotherapy-discussions across neuro-oncology and non-neuro-oncology experts in cancer immunology. Meeting Report from the 2019 SNO Immuno-Oncology Think Tank. *Neuro. Oncol.* **23**, 356–375 (2021).
18. Klemm, F. *et al.* Interrogation of the Microenvironmental Landscape in Brain Tumors Reveals Disease-Specific Alterations of Immune Cells. *Cell* **181**, 1643-1660.e17 (2020).
19. Medikonda, R. *et al.* Synergy between glutamate modulation and anti-programmed cell death protein 1 immunotherapy for glioblastoma. *J. Neurosurg.* **136**, 379–388 (2022).
20. Heuer, S. *et al.* PerSurge (NOA-30) phase II trial of perampanel treatment around surgery in patients with progressive glioblastoma. *BMC Cancer* **24**, 135 (2024).
21. Tobochnik, S. *et al.* Pilot trial of perampanel on peritumoral hyperexcitability and clinical outcomes in newly diagnosed high-grade glioma. *medRxiv* (2024)
doi:10.1101/2024.04.11.24305666.
22. Genoud, V. *et al.* Responsiveness to anti-PD-1 and anti-CTLA-4 immune checkpoint blockade in SB28 and GL261 mouse glioma models. *OncoImmunology* vol. 7 e1501137 Preprint at <https://doi.org/10.1080/2162402x.2018.1501137> (2018).
23. Simonds, E. F. *et al.* Deep immune profiling reveals targetable mechanisms of immune evasion in immune checkpoint inhibitor-refractory glioblastoma. *Journal for ImmunoTherapy of Cancer* vol. 9 e002181 Preprint at <https://doi.org/10.1136/jitc-2020-002181> (2021).
24. Kosaka, A., Ohkuri, T. & Okada, H. Combination of an agonistic anti-CD40 monoclonal antibody and the COX-2 inhibitor celecoxib induces anti-glioma effects by promotion of type-1 immunity in myeloid cells and T-cells. *Cancer Immunol. Immunother.* **63**, 847–857 (2014).
25. Genoud, V. *et al.* Treating ICB-resistant glioma with anti-CD40 and mitotic spindle checkpoint controller BAL101553 (lisavanbulin). *JCI Insight* **6**, (2021).
26. Letchuman, V. *et al.* Syngeneic murine glioblastoma models: reactionary immune changes and immunotherapy intervention outcomes. *Neurosurg. Focus* **52**, E5 (2022).

27. Kay, K. E. *et al.* Tumor cell-derived spermidine promotes a protumorigenic immune microenvironment in glioblastoma via CD8⁺ T cell inhibition. *J. Clin. Invest.* **135**, (2024).
28. Maeoka, R. *et al.* Local administration of shikonin improved the overall survival in orthotopic murine glioblastoma models with temozolomide resistance. *Biomed. Pharmacother.* **166**, 115296 (2023).
29. Lee, J. *et al.* Sex-biased T-cell exhaustion drives differential immune responses in glioblastoma. *Cancer Discov.* **13**, 2090–2105 (2023).
30. Watson, D. C. *et al.* GAP43-dependent mitochondria transfer from astrocytes enhances glioblastoma tumorigenicity. *Nat Cancer* **4**, 648–664 (2023).
31. Liu, S. J. *et al.* In vivo perturb-seq of cancer and microenvironment cells dissects oncologic drivers and radiotherapy responses in glioblastoma. *Genome Biol.* **25**, (2024).
32. Pan, S. *et al.* Multiplatform molecular profiling and functional genomic screens identify prognostic signatures and mechanisms underlying MEK inhibitor response in somatic *NF1* mutant glioblastoma. *bioRxiv* 2024.07.01.601334 (2024)
doi:10.1101/2024.07.01.601334.
33. Lad, M. *et al.* Glioblastoma induces the recruitment and differentiation of dendritic-like “hybrid” neutrophils from skull bone marrow. *Cancer Cell* **42**, 1549-1569.e16 (2024).
34. Lawrence, M. S. *et al.* Mutational heterogeneity in cancer and the search for new cancer-associated genes. *Nature* **499**, 214–218 (2013).

Dear Editor and Reviewers:

We sincerely appreciate the reviewers' thoughtful and constructive feedback on our revised manuscript, "Glioma-neuronal circuit remodeling induces regional immunosuppression." We are grateful for the time and effort dedicated to reviewing our work.

Reviewer #4 (Remarks to the Author):

The author's point-to-point response and the revised manuscript have thoroughly addressed all my concerns. The data is of high quality and enhances our understanding of the interactions between the nervous system and the glioma immune microenvironment. It is an excellent and highly valuable study. I recommend this manuscript for publication.

Response: We thank the reviewer for the careful evaluation of our responses and the revised manuscript. We are especially grateful for the encouraging comments and strong endorsement for publication.